# SIGNING THE SUPERMASK: KEEP, HIDE, INVERT

**Nils Koster**
Department of Applied Econometrics
Karlsruhe Institute of Technology
nils.koster@kit.edu

**Oliver Grothe**
Department of Analytics and Statistics
Karlsruhe Institute of Technology
oliver.grothe@kit.edu

**Achim Rettinger**
Department of Computational Linguistics and Digital Humanities
Trier University
rettinger@uni-trier.de

## ABSTRACT

The exponential growth in numbers of parameters of neural networks over the past years has been accompanied by an increase in performance across several fields. However, due to their sheer size, the networks not only became difficult to interpret but also problematic to train and use in real-world applications, since hardware requirements increased accordingly. Tackling both issues, we present a novel approach that either drops a neural network's initial weights or inverts their respective sign. Put simply, a network is trained by weight selection and inversion without changing their absolute values. Our contribution extends previous work on masking by additionally sign-inverting the initial weights and follows the findings of the Lottery Ticket Hypothesis. Through this extension and adaptations of initialization methods, we achieve a pruning rate of up to 99%, while still matching or exceeding the performance of various baseline and previous models. Our approach has two main advantages. First, and most notable, signed Supermask models drastically simplify a model's structure, while still performing well on given tasks. Second, by reducing the neural network to its very foundation, we gain insights into which weights matter for performance. The code is available here.

## 1 INTRODUCTION

In recent years neural networks have established themselves as versatile problem solvers far beyond the field of machine learning. However, the performance increases achieved are obtained by an even larger growth in their size.This not only leads to growing energy costs (Strubell et al., 2020), but also complicates interpretability considerably. There are different approaches to reduce various aspects of complexity within a neural network, which broadly can be categorized either as pruning, i.e., the technique of trimming down neural networks, or as low-precision training, i.e., representing the weights by low precision numbers. Pruning has shown to significantly reduce the size of a neural network (in terms of "active" weights) without affecting performance considerably, once the training phase has ended. Compared to the original network, the pruned network is sparse, since many of the original, "useless" weights are set to zero. In contrast, replacing weights by low precision numbers, reduces memory size, but does not reveal a sparse, likely easier to interpret network structure.

Many approaches of pruning (e.g., Janowsky (1989); LeCun et al. (1990); Han et al. (2015); Li et al. (2016b); Molchanov et al. (2019) among others) exist to identify "useless" weights. This raises the question of why not train a smaller network from scratch, since the training of a large network is the computationally most expensive step and pruning is another additional step. Frankle & Carbin (2018) present a possible solution to this question by formulating the Lottery Ticket Hypothesis (LTH), which states that any densely connected neural network contains smaller subnetworks that, when trained in isolation, perform just as well or even better than the original network.
They are able to identify such subnetworks (*winning tickets*) by iteratively pruning the network based on the magnitude of each weight. For deeper architectures, Frankle et al. (2019) show, that pruning after a few epochs of normal training works significantly better over pruning at initialization.

Zhou et al. (2019) follow the seminal idea of the LTH and are able to train neural networks by only selecting *untrained* weights (i.e., weights are frozen after initialization), a concept they call *Supermasks*. In other words, they find a smaller subnetwork during training without adjusting the weights themselves. Although this approach did not match the performance of their baselines and the pruning rate is inconsistent, it revealed a startling insight: weight values do not seem to be as important as the connection itself; a single, well-initialized value for each layer is sufficient. Ramanujan et al. (2020) further develop Supermasks by modifying the way masks are calculated, which leads to significant performance improvements compared to Zhou et al. (2019). However, the number of parameters is still high. Recently, Chijiwa et al. (2021) modified the approach of Ramanujan et al. (2020) by randomizing the scores, but prune their networks only up to 60%.

In this paper, we propose a technique called **signed Supermask**, a natural extension of Zhou et al. (2019) and Ramanujan et al. (2020). We not only determine the importance of a weight by masking, but also learn the respective sign of a weight. Signed Supermasks aim at very sparse structures and are able to uncover subnetworks with far fewer parameters in the range of 0.5% to 4% of the original network requiring little additional computational effort without sacrificing performance.

This differs substantially from low precision training. In its most extreme form, low-precision training quantizes the weight values of a neural network to three constants, including zero or two constants, excluding zero. There, binarized neural networks (BNN) (Courbariaux et al., 2015) reduce complexity but not the size of the networks in terms of sparsity of weights. Ternary neural networks (TNN), introduced by Li et al. (2016a) allow in principle for sparse subnetworks, however this literature focuses almost exclusively on optimizing computational costs while maintaining predictive performance. For TNNs to reduce computational complexity, sparsity (and interpretability) is of no importance, as they are focused on reducing the computational footprint only. The literature presents diverse approaches, for example Shayer et al. (2017) utilize the local reparameterization trick (Kingma et al., 2015) to learn ternarized weights stochastically, Zhu et al. (2016) ternarize the weights but scale them layer-wise by two learned real-valued scalars and Alemdar et al. (2016) employ a teacher-student approach. Deng & Zhang (2020) train their networks normally with an additional regularization term in the loss function and only ternarize at the end of training by rounding. To the best of our knowledge, this is the only work on TNNs also reporting on sparsity. More recently, Diffenderfer & Kailkhura (2021) combined the edge-popup algorithm by Ramanujan et al. (2020) and the binarization of weights and achieve good results.
Thus, although TNNs and Supermasks in general appear similar at first glance, their goals are different: while the former attempt to reduce computational complexity, the latter attempt to find the smallest possible subnetworks within a neural network to better understand neural networks in general and work towards facilitating interpretability.

This paper takes the Supermasks perspective and our experiments show, that signing the Supermask matches or outperforms baselines and state-of-the-art approaches on Supermasks and leads to very sparse representations. A convenient side effect of signed Supermask is the ternarization of weights, with implied reduced memory requirements (Li et al., 2016a) and further significant speedup in inference (Hidayetoglu et al., 2020; Brasoveanu et al., 2020). In summary, signed Supermasks provide two major advantages. First, the network structure is simplified while maintaining or improving the performance compared to their dense counterpart. Second, the reduction facilitates a better understanding of the inner mechanics of neural networks. Based on that, we might be able to build smaller but equally powerful models a priori. Additionally, once trained, signed Supermask models can be stored more efficiently and have the potential for faster inference.

## 2 SIGNED SUPERMASK

The idea of a signed Supermask is simple, but elegant: the network may not only switch off a given weight, but may also flip its sign if deemed beneficial. We do this by multiplying each weight with a mask $m \in \{-1, 0, 1\}$, in contrast to $m \in \{0, 1\}$ as in previous work (Zhou et al., 2019; Ramanujan et al., 2020).

For example, if a single weight is initialized with a negative sign but the performance of the network would increase if it was positive, the signed Supermask has the additional degree of freedom to flip the weight's sign. This step decreases the impact of random initialization, which matters specifically for a *signed constant initialization*, as introduced by Zhou et al. (2019) in the context of Supermasks.

Despite including -1 in $m$, we have no additional information to store for the final sparse weight matrices (i.e., the final mask multiplied with the weight matrix), since it already includes the weight value, the sign and the zero. In this regard, the additional flexibility in the training phase comes for free.

Note that signed Supermasks are applicable to any neural network architecture. By masking, we neither alter the information flow, nor any architecture specific properties. For the sake of brevity, however, we focus in the following on the application on feed-forward neural networks only.

The general setup is as follows: Let $\boldsymbol{W}_l \in \mathbb{R}^{n_l \times n_{l-1}}$ denote the frozen weight matrix of layer $l \in \{1, \ldots, L\}$ of a feed-forward neural network with $L$ layers and $n_l$ denoting the number of neurons in layer $l$. We initialize $\boldsymbol{W}_l \sim \mathcal{D}_w$. Let further $\boldsymbol{M}_l \in \mathbb{R}^{n_l \times n_{l-1}}$, $\boldsymbol{M}_l \sim \mathcal{D}_m$ be a learnable, real-valued matrix that is transformed to values $\{-1, 0, 1\}$ by a element-wise function $g$, where

$$g(\boldsymbol{M}) = \begin{cases} -1, & \boldsymbol{M} \leq \tau_n \\ 0, & \tau_n < \boldsymbol{M} < \tau_p \\ 1, & \boldsymbol{M} \geq \tau_p \end{cases} \tag{1}$$

for some thresholds $\tau_n$ and $\tau_p$ as well as appropriate distributions $\mathcal{D}_w$ and $\mathcal{D}_m$ that will be specified below. We now define the signed Supermask as $\bar{M} := g(\boldsymbol{M})$, i.e., a Supermask as in Zhou et al. (2019) with the additional degree of freedom of a negative sign. Further note that, compared to Zhou et al. (2019), the function $g(\cdot)$ is deterministic. In order to make full use of the *signed* Supermask, we utilize the ELU (**E**xponential **L**inear **U**nit) activation function (Clevert et al., 2015) with hyperparameter $\alpha$. Hereafter, we always set $\alpha = 1$. Then, with $\odot$ symbolizing the Hadamard product, each layer $l$'s output $\boldsymbol{o}_l$ with $\boldsymbol{z}_l = (\boldsymbol{W}_l \odot \bar{\boldsymbol{M}}_l)^T \boldsymbol{o}_{l-1}$ is computed as follows:

$$\boldsymbol{o}_l = \begin{cases} \boldsymbol{z}_l, & \boldsymbol{z}_l > 0 \\ e^{\boldsymbol{z}_l} - 1, & \boldsymbol{z}_l \leq 0 \end{cases} \tag{2}$$

The parameters $\tau_{n,p}$ are hyperparameters that need to be tuned: the larger they are in absolute value, the more weights are pruned at initialization and the larger values in $\boldsymbol{M}$ need to grow during the course of training to "activate" its responding weight. The values of $\tau_{n,p}$ might differ depending on the task and network architecture at hand. The gradient is estimated using the straight-through estimator (Hinton, 2012; Bengio et al., 2013) as in previous work (Zhou et al., 2019; Ramanujan et al., 2020). That is, in the backward-pass, we regard the threshold function $g(\cdot)$ as the identity function. Thus, the gradient is going "straight through" $g(\cdot)$.

Zhou et al. (2019) generate their masks by sampling the entries from Bernoulli distributions with learnable Bernoulli probabilities. Their $\boldsymbol{M}$ therefore resembles the probability, that weight $w_{ij}$ in some layer $l$ is active. Sampling from a Bernoulli distribution can also be interpreted as a stochastic threshold function. Our **fixed threshold** approach follows the same intuition with the difference of not using a stochastic sampling method. That is, we define two thresholds $\tau_n$ and $\tau_p$ after the initialization phase which remain fixed during the course of training in contrast to Ramanujan et al. (2020) who update $\tau_n$ and $\tau_p$ after each epoch such that the pruning rate stays constant. Having fixed thresholds has three main advantages which are in line with the argumentation of Zhou et al. (2019): first, the network can choose the optimal number of remaining weights. Second, it can choose the optimal distribution of 1's and $-1$'s. Third, this approach is fast, as we only have to compute $\tau_n$ and $\tau_p$ once. Moreover, this method, with some modifications, is frequently used in the TNN literature, which underlines its legitimacy. As a result, an exact pruning rate cannot be determined a priori, but is learned. However, we show in Appendix F, that the pruning rate can be indirectly influenced by adjusting other hyperparameters.

## 2.1 THOUGHTS ON INITIALIZATION

Signed Supermasks do not alter the value of any weight during training. Therefore, it is especially important to initialize weights appropriately. Common weight initialization methods such as He (He et al., 2015) or Xavier (Glorot & Bengio, 2010) neither take masking into account nor the ELU activation function.

**Weights**   We propose a weight initialization scheme that considers both aforementioned alterations, namely, the signed masking process and a differing activation function. In simple terms, since each

layer holds fewer weights, we can scale the weight value to counteract masking. Considering that the goal of initialization methods is to keep the variance in the forward- and backward-pass constant, we suggest to use the following variance:

$$\text{Var}[\boldsymbol{W}_l] = \frac{1.5}{n_l(1 - p_0^l)} \tag{3}$$

with fan-out $n_l$ and layer-wise equal, initial probabilities $p_0^l = P(\bar{\boldsymbol{M}}_{ij}^l = 0)$ for layer $l \in \{1, ..., L\}$ and $\forall i \in 1, \ldots, n_l, j \in 1, \ldots, n_{l-1}$. A detailed formal motivation of Equation 3 under specified assumptions can be found in Appendix A. The result is still valid without masking, i.e., $p_0^l = 0$. He et al. (2015) come to a similar conclusion for their Parametric ReLU (PReLU) activation function. Henceforth, we refer to this initialization method as **ELUS** (**ELU S**caled) with masking or simply **ELU** in case of no masking. Notice that ELUS is simply a scaled version of He initialization (He et al., 2015), where the scalar is only dependent on the hyperparameter $\alpha$ of the ELU activation function which is known before training (and set to 1 in this work).

**Mask**    Our assumptions and observations regarding the initialization of the real-valued mask $\boldsymbol{M}$ are as follows: We assume that $\mathcal{D}_m$ is symmetric around zero, a standard assumption for common initialization distributions. Given the properties of $\boldsymbol{M}$, $\bar{\boldsymbol{M}}$ and the gradient straight-through estimator, it is advantageous if the values in $\bar{\boldsymbol{M}}$ and $\boldsymbol{W}$ are roughly of the same magnitude, such that the gradient, which is independent of $\boldsymbol{M}$, is neither too small nor too large. In other words, $\mathcal{D}_m$ should be chosen such that it is symmetrical around zero and has a similar variance as $\mathcal{D}_w$. In the case of mask initialization, we advice against a signed constant approach but recommend a normal or uniform distribution instead.

## 3    EMPIRICAL ASSESSMENT

The following paragraphs present the experiments conducted in order to assess the validity and effectiveness of fixed threshold signed Supermasks. The code for the experiments can be found here.

**Experimental Setup**    To ensure comparability, we utilize the same experimental setup as Zhou et al. (2019) and Frankle & Carbin (2018) i.e., models FCN, Conv2, Conv4 and Conv6 (VGG-like architectures (Simonyan & Zisserman, 2014)) with the additional Conv8 model of Ramanujan et al. (2020) without batch normalization (Ioffe & Szegedy, 2015) and dropout (Srivastava et al., 2014). Since we want to study the functionality of signed Supermasks in particular, any extra functionalities would disturb from the essential topic even though it might bump performance. Details about the aforementioned model architectures can be found in Table 7 in Appendix B. Furthermore, we study signed Supermasks on ResNet20, ResNet56 and ResNet110 (He et al., 2016) with the exception of replacing ReLU with ELU (similar to Shah et al. (2016)). The batch normalization layers in the ResNet models are frozen for signed Supermask training and are only used because its standard practice. For all experiments the same architectures as the respective signed Supermask models but trained conventionally, act as baselines.

The fully-connected (FCN) model is trained on MNIST (LeCun et al., 2010) and all CNNs plus ResNet20 on CIFAR-10 (Krizhevsky, 2009). We only apply per image standardization on both of those datasets. ResNet56 and ResNet110 are trained on CIFAR-100 (Krizhevsky, 2009), where we apply per image standardization, pad the image with 4 pixels on each side and randomly crop a 32x32 image out of the original or its horizontally flipped counterpart.

We ran each experiment 50 times (ResNet56/ResNet110: 3 runs) with the same setup but different weights and masks drawn from the respectively same specified distribution. This ensures that our experimental results are not affected by the randomness of the initialization step. To obtain an intuition of the scale of memory reduction, we provide a simple metric by comparing the average network size in bytes stored in `numpy` arrays and scipy's `csr_matrix`[1] sparse matrices after training. For matrices with more than 2 dimensions, we reshape the matrix to two dimensions. This does not give an exact result but provides a good heuristic. Since we attempt to reduce complexity of a neural network, we also provide the average time needed to train a model for a single epoch to further assess practicability.

---

[1]The choice of `csr` is arbitrary. The purpose of this metric is not to show how we can most efficiently store the networks but to give an intuition on added value.

Table 1: FCN: Test accuracy and training time. We report the mean, 5% and 95% quantiles of 50 runs for each initialization method as well as the results of Zhou et al. (2019) and Ramanujan et al. (2020). Mean training time per epoch is reported in the last column.

| | Init | Test Accuracy [%] | Rem. Weights [%] | TT / Epoch [s] |
|---|---|---|---|---|
| *Baseline* | He | 97.40 [97.3, 97.5] | 100 | 1.23 |
| | ELU | 97.42 [97.3, 97.5] | 100 | 1.21 |
| | Xavier | 97.43 [97.4, 97.5] | 100 | 1.21 |
| *Signed Supermask* | He | 97.12 [97.0, 97.3] | 4.93 [4.9,5.0] | 1.36 |
| | ELUS | 97.48 [97.3, 97.7] | 3.77 [3.7,3.8] | 1.27 |
| | Xavier | 96.89 [96.8, 97.0] | 5.51 [5.5,5.6] | 1.32 |
| *Zhou et al.* | Xavier (S.C.) | 98.0 | 11 - 93 | - |

**Initialization**     To initialize the weights of the signed Supermask models, we use signed constants in combination with Xavier, He and ELUS. Thus, the respective weight value has a 50% probability of being negative, where the absolute value of the constant is influenced by the respective initialization method. We initialize the baseline models with the same method within a uniform distribution as the respective signed Supermask model. The idea of using only one (well-chosen) signed constant is particularly compelling from the perspective of complexity reduction. For the ELUS initialization, we scale a He initialization by $\sqrt{3}$ for FCN and Conv architectures and $\sqrt{1.5}$ for our ResNets, which returned best results in the preliminary testing phase. All masks are initialized with Xavier uniform initialization. Results of experiments with different mask initializations and distributions can be found in Appendix C.

## 3.1    FULLY-CONNECTED NETWORKS

This section evaluates the performance of the fully-connected neural network (FCN) signed Supermask model. Hyperparameters used for training the FCN are listed in Tables 8 and 10 in Appendix B. Note that compared to Zhou et al. (2019), the parameter choices are in line with standard practice. Table 1 summarizes the average test accuracy, remaining weights and training time per epoch with the respective 5% and 95% quantiles for both baseline and signed Supermask models with the different weight initializations. All models achieve similar test accuracy. Best performing is the ELUS signed Supermask model. Furthermore, all signed Supermask models have a wider confidence interval, indicating higher variance. In contrast, confidence intervals for the ratio of remaining weights are very small, demonstrating constant pruning rates throughout all runs. Furthermore, ELUS' ratio of remaining weights of 3.77% is considerably lower than for the other two initializations. This indicates a superior initialization as the network is able to achieve a higher performance with fewer weights remaining. For another angle of observation, Figure 5 in Appendix C visualizes the average ratio of remaining weights over the course of training for each weight initialization. Regarding efficiency, ELUS signed Supermask models require about 5% more training time. However, the obtained compression rate for the ELUS model after training is 92.01% on average, which makes up for the additional training effort. Although the performance of the ELUS signed Supermask is trailing 0.5pp behind the FCN model of Zhou et al. (2019), we argue that considering the pruning rate (they state the pruning rate only imprecisely, ranging from 7% to 89%) and performance, signed Supermasks are advantageous.

To assess how the model interprets a given dataset, we examine the first masking layer of the network, visualized in Figure 1. The top image represents an input vector (i.e., a flattened MNIST sample), with each vertical line representing one pixel. The bottom shows the mask of the first layer of a randomly selected trained ELUS FCN model. The signed Supermask has learned which of the input nodes are more important than others. Those that match a background pixel are completely masked, while the input nodes that receive actual information are allowed to let the information pass through. Thus, the first layer acts as a filter and feature selector. We further analyze the behavior of the masks and draw comparisons to binary Supermasks in Appendix C.

The results so far show the potential of signed Supermasks in relatively small architectures. Not only does the ELUS model outperform the respective baselines with only 3.8% of the original weights, it

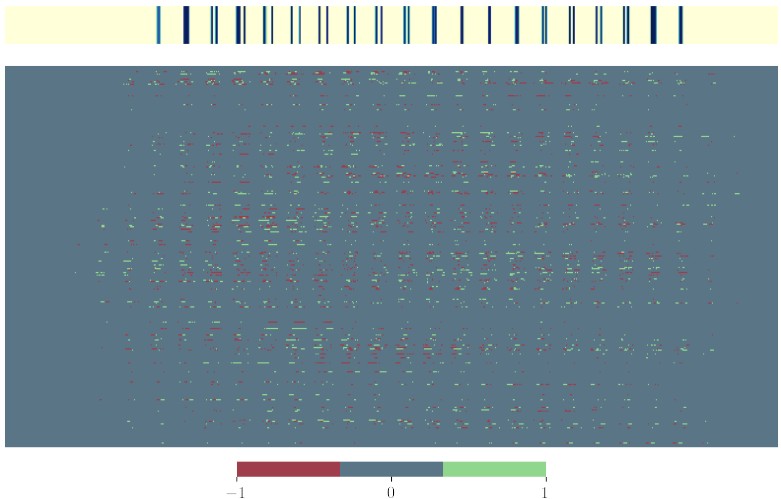

Figure 1: FCN Signed Supermask: The upper figure shows a randomly chosen (in this case the digit "8") flattened MNIST sample. Each line represents a single pixel where non-yellow lines represent the pixels containing a part of the handwritten digit. The lower figure visualizes the first mask of a randomly picked and trained ELUS FCN model. We can clearly see the alignment of the "digit-pixels" in the MNIST sample and the non-zero mask elements.

also saves 92% of memory once trained. Furthermore, we gain insights into layer interpretability as pruning is not evenly distributed across the network.

## 3.2 Convolutional Networks

Here, we present the main experimental results of convolutional neural networks (CNN) signed Supermasks. Overfitting was a major issue in the development, especially for the baseline models. To counter that, the baseline models are only trained for 50 epochs. In contrast, signed Supermask models were not as prone to overfitting. However, once we anticipated overfitting tendencies of Conv2, we reduced its learning rate compared to the other CNN models. Apart from those alterations, the hyperparameter choices are the same throughout all CNN architectures. All models are optimized with SGD with weight decay. Learning rate for Conv models are exponentially reduced over the course of training. Tables 8 and 10 in Appendix B summarize all training parameters. We have found in initial experiments that $\pm0.01$ is a solid choice for the threshold parameters $\tau_n$ and $\tau_p$. This corresponds to an initial overall pruning rate of about 11% to 30%.Thus, larger layers initially experience a slightly higher pruning rate than smaller layers, following the intuition that weights in smaller layers impact the output of the network to a greater extent than weights in larger layers. Table 2 presents test accuracy for our baselines, the models of Zhou et al. (2019) and Ramanujan et al. (2020) and signed Supermasks as well as remaining weights (i.e. 1 - pruning rate) for the latter three. For a broader comparison to literature, see Table 14 in Appendix C. While we were able to reproduce the results of Zhou et al. (2019), this was not the case for Ramanujan et al. (2020) which not only delivered worse results than reported but also demanded a multiple of the training time. The interested reader can find additional results, experiments on different weight and initialization methods as well as investigations on extreme pruning rates in Appendices C D, E and F. First, apart from Conv2, all CNN models significantly outperform their respective baselines. Conv2 reaches similar accuracy to its baseline, the difference only being 0.4pp. The more our models grow in depth, the greater the benefit of signed Supermasks over the baselines. When considering the ratio of remaining weights, a likely reason for the slight performance glitch of Conv2 becomes apparent: it utilizes only 0.6% of the original weights. In other words, Conv2 reaches similar performance with 99.4% of the original weights unused. Conv4, the smallest model in terms of absolute parameter count, keeps the highest percentage of weights in relation to the original count. As the networks grow in size and depth, we can once again note an increase in the pruning rate. Except for Conv6, there is hardly any variance at all in this metric, indicating a robust behavior.

Table 2: Conv: Test accuracy and remaining weights. We report the mean, 5% and 95% quantiles of 50 runs for the ELUS signed Supermasks and baselines. The respective best results of Zhou et al. (2019), Ramanujan et al. (2020) (abbreviated "Ram") and Diffenderfer & Kailkhura (2021) (abbreviated "Diff") are shown as well. Evidently, signed Supermasks outperform the previous literature on Supermasks on all CNN architectures, taking our extreme pruning rates of at least 97% into account. Apart from Conv2, we also gain higher performance than the baseline models.

| | Accuracy [%] | | | | | Rem. Weights [%] | | | |
| | Baseline | Zhou | Ram. | Diff. | Sig. Supermask | Zhou | Ram. | Diff. | Sig. Supermask |
|---|---|---|---|---|---|---|---|---|---|
| Conv2 | 68.79 [68.4, 69.2] | 66.0 | 65 | 70 | 68.37 [67.7, 69.0] | 11 - 93 | 10 | 10 | 0.60 [0.58, 0.62] |
| Conv4 | 74.50 [74.7, 75.3] | 72.5 | 74 | 79 | 77.40 [76.7, 78.3] | 11 - 93 | 10 | 10 | 2.91 [2.9, 3.0] |
| Conv6 | 75.91 [75.4, 76.4] | 76.5 | 77 | 82 | 79.17 [78.1, 80.5] | 11 - 93 | 10 | 10 | 2.36 [1.9, 2.6] |
| Conv8 | 76.24 [75.1, 77.2] | - | 70 | 85 | 80.91 [79.9, 81.7] | - | 10 | 10 | 1.17 [1.1, 1.2] |

Table 3: Conv Signed Supermask: Additional training time (abbreviated "TT" below) and compression rate compared to the respective baselines. Signed Supermask CNN models train roughly 6 to 10% longer, however, once trained, their required memory can be reduced by at least 93.8%.

| | Add. TT/Epoch [%] | Compression Rate [%] |
|---|---|---|
| Conv2 | 10.14 | 98.41 |
| Conv4 | 6.02 | 93.82 |
| Conv6 | 6.25 | 95.07 |
| Conv8 | 6.00 | 97.6 |

In terms of efficiency, we see in Table 3 that the signed Supermask models require 6% to 10% more training time, which can be accounted to the additional calculations necessary. However, once the networks are trained the required memory can, on average, be compressed by 94% to 98%. Signed Supermasks outperform not only their respective baselines but also the methods proposed in the related work. Compared to Zhou et al. (2019), our Conv 2, Conv4 and Conv 6 models achieve a higher performance of at least 2pp. Diffenderfer & Kailkhura (2021) do not report performance over a 90% pruning rate, however, their models achieve higher accuracy than ours. We account this to their lower pruning rate. Ramanujan et al. (2020) do not report model performance below a pruning rate of 90% as well. Indeed, in this case, all our Conv models outperform their equivalent models by 3 to 10 pp, while uncovering much smaller subnetworks.

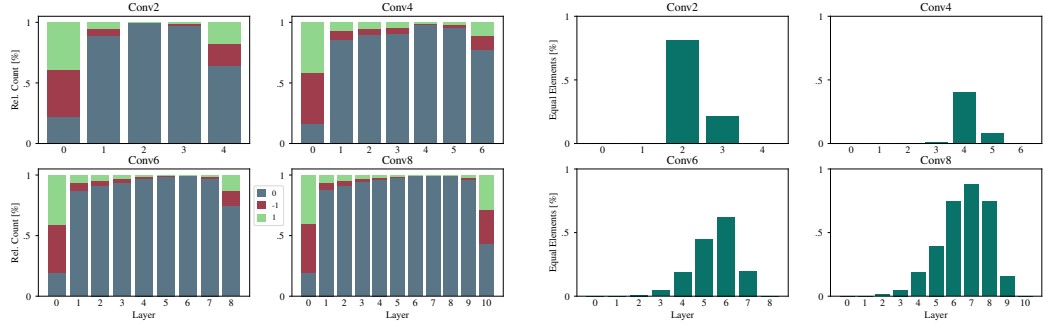

(a) Average ELUS mask distribution. We can observe, that sparsity is very high in very wide layers, indicating that only few of those weights are needed.

(b) Equality of all 50 final ELUS masks. Masks show high element-wise equality where sparsity is highest.

Figure 2: Conv Signed Supermask: Mask distribution and equality.

Figure 2a helps to interpret the masks by visualizing the average layer-wise mask distribution of each architecture. It can be clearly seen that the first layers are least affected by pruning, while all other layers reach a significantly higher pruning rate of over 99%. As the layer size increases, the pruning rate also increases and peaks in the largest layer of the network. The output layers are also relatively sparse, again indicating proper initialization. This finding confirms our previous assumption that

larger layers contain far fewer weights that are essential to the network. The relative element-wise equality of the final masks over all 50 runs for all layers and architectures is shown in Figure 2b. This provides an intuition about the similarity of the masks and therefore subnetworks through different runs. We observe that pairwise equality is strongly correlated with the size and sparsity of a layer: the larger, the sparser, the more alike. To some degree this is expected as very sparse layers are alike by chance. However, measuring similarity in matrices is a very difficult task (for an overview on this literature from a graph-based perspective see, e.g. Emmert-Streib et al. (2016)), which is why we provide visualizations in Figure 9 in Appendix C that emphasize especially the similarity in structure. This shows, that signed Supermasks repeatedly find strikingly similar subnetworks for different runs, meaning that it might be possible to identify certain neurons that are more important than others and hence better understand neural networks in general.

## 3.3 RESIDUAL NETWORKS

To further investigate the effectiveness of signed Supermasks, we turn our attention towards Residual networks which allow for deeper architectures. The investigated ResNet consist of 9,27,54 residual blocks, each in equal parts with 16, 32 and 64 filters, respectively. This sums up to a comparatively small parameter count of roughly 0.25M, 0.84M and 1.7M. As hypothesized above, signed Supermasks work better in deeper architectures that contain enough parameters to find a good subnetwork. To investigate this, we multiply the number of filters by 1.5, 2, 2.5 and 3 to create wider variants of ResNet20. Note that even the largest of those variants is smaller than any of the studied VGG-like architectures above, suggesting less overparameterization, especially for the more complex CIFAR-100 dataset. Hyperparameters for baseline and signed Supermask models are depicted in Tables 9 and 11 in Appendix B. Compared to the other investigated architectures, we only reduce the learning rate once the train loss hits a plateau. Furthermore, since there is no large deviation between layer sizes, we set the threshold parameters $\tau_n$ and $\tau_p$ such that only 25% of the weights are left for ResNet20 and 15% for ResNet56 and ResNet110.

Table 4 shows the test accuracy and remaining weights of both baselines and signed Supermasks trained on CIFAR-10. Notably, signed Supermask always trail 2 to 3 pp behind their respective baseline. However, once the signed Supermask model gets wide enough it can reach the performance of the original ResNet20 with only a third of the parameter count. This finding is partially in line with Ramanujan et al. (2020) and their scaling of ResNet50. As expected, the pruning rate increases as the network grows in depth. We suspect that weight count could be further dropped in the deep architectures by adapting the weight decay parameter, for the sake of clarity however, we decided to keep all learning parameters the same. A possible reason for the slightly lower performance of signed Supermasks might be the frozen batch normalization layers, which would benefit learning greatly. We investigate this aspect in Section 3.3.1. The performance and remaining weights of ResNet56 and

Table 4: ResNet20: Test accuracy and remaining weights on CIFAR-10. We report the mean, 5% and 95% quantiles for the ELUS signed Supermasks and ELU baselines. ResNet20 signed Supermasks are not able reach the performance of the baselines. However, the wider the ResNet20s become, the closer they get to the respective baseline and the higher the pruning rate. Furthermore, with a multiplier of 2.5 and 3, the signed Supermasks reach the performance of the original baseline with a third of the original weights.

|  | Accuracy [%] | | Rem. Weights | |
|  | Baseline | Sig. Supermask | Baseline | Sig. Supermask |
| --- | --- | --- | --- | --- |
| ResNet20 | 84.91 [84.39, 85.56] | 81.68 [81.03, 82.62] | 248624 | 53530 (21.13%) |
| ResNet20x1.5 | 86.18 [85.59, 86.74] | 83.76 [83.01, 84.31] | 558600 | 64223 (11.93%) |
| ResNet20x2.0 | 86.80 [86.39, 87.21] | 84.42 [83.67, 85.26] | 992352 | 77280 (7.69%) |
| ResNet20x2.5 | 87.08 [86.72, 87.42] | 84.71 [84.26, 85.32] | 1549880 | 84446 (5.42%) |
| ResNet20x3.0 | 87.32 [87.00, 87.67] | 84.89 [84.36, 85.48] | 2231184 | 91798 (4.06%) |

ResNet110 on CIFAR-100 are depicted in Table 5. Signed Supermasks show, that they successfully learn from the dataset, even in deep and relatively small environments. However, they cannot reach the same test accuracy as their respective baseline. This can be partially explained by the relatively

small original network sizes. Furthermore, as hypothesized above, batch normalization is of great importance especially in deep architectures like ResNet110, which thereby particularly suffered.

Table 5: ResNet56/110: Test accuracy and remaining weights on CIFAR-100. We report the mean, estimated 5% and 95% quantiles for the ELUS signed Supermasks and ELU baselines as well as the utilized weights. Signed Supermask models trail behind their baseline, however, utilizing a maximum of 29% of the original weights.

|  | Accuracy [%] | | Rem. Weights | |
|  | Baseline | Sig. Supermask | Baseline | Sig. Supermask |
|---|---|---|---|---|
| ResNet56 | 68.04 [67.84, 68.24] | 60.01 [59.51, 60.52] | 834992 | 245116 (29.39%) |
| ResNet110 | 62.70 [54.80, 68.61] | 46.42 [46.31, 46.51] | 1668272 | 346757 (20.64%) |

### 3.3.1 THE ROLE OF BATCH NORMALIZATION

ResNets excel compared to conventional CNNs because of their depth. A big role in being able to learn very deep architectures is batch normalization. Here, we want to study the influence of batch normalization on the ResNet signed Supermask architectures. For those experiments, we explicitly turn on the batch normalization layers in the respective architecture. This parts a bit from the minimalist idea of Supermasks in general, however, weights are still frozen at initialization and this step might be necessary for deep architectures.
Table 6 reports the results of the baseline models (as displayed above) and the signed Supermask with learnable batch normalization layers. Compared to the pure signed Supermask models, batch normalization helps to push performance while increasing pruning rate for all models. However, they are unable to reach the performance of the baselines. Nevertheless, taking the remaining weights into account, the predictive performance impresses. More research is needed to investigate the performance gap. Besides this, the results underline that batch normalization helps signed Supermasks to gain better performance in very deep architectures while utilizing fewer weights.

Table 6: ResNet BN: Test accuracy and remaining weights on CIFAR-10 for ResNet 20 and CIFAR-100 for ResNets 56 and 110. We report the mean, estimated 5% and 95% quantiles for the ELUS signed Supermasks and ELU baselines as well as the utilized weights.

|  | Accuracy [%] | | Rem. Weights | |
|  | Baseline | Sig. Supermask BN | Baseline | Sig. Supermask BN |
|---|---|---|---|---|
| ResNet20 BN | 84.91 [84.39, 85.56] | 83.38 [82.87, 83.92] | 248624 | 37992 (15.54%) |
| ResNet56 BN | 68.04 [67.84, 68.24] | 64.24 [63.90, 64.58] | 834992 | 231769 (27.59%) |
| ResNet110 BN | 62.70 [54.80, 68.61] | 53.67 [53.63, 53.72] | 1668272 | 64663 (3.86%) |

## 4 SUMMARY

In this work, we introduced the concept of **signed Supermasks**. Extending the original work on Supermasks (Zhou et al., 2019), we include the sign in the masking process and utilize a simple step-function to calculate the effective masks. Additionally, we gave theoretical and experimental justification for an adapted weight initialization scheme to the masking process, called **ELUS**.
Our experiments on MNIST and CIFAR-10 show, that signed Supermask models at best hold as little as 0.6% of the original weights, while performing on an equal or superior level compared to our baselines and current literature (Zhou et al., 2019; Ramanujan et al., 2020). As a side effect, the trained signed Supermask models' memory footprint can be compressed by up to 98%. Further analyses on CIFAR-100 with ResNet56 and ResNet110 suggest that signed Supermasks are generally able to work on complex datasets and architectures as well.

ETHICS STATEMENT

The authors cannot envision any socioeconomic or ethical downsides that are specific to signed Supermasks but not neural nets or machine learning in general. On the contrary, we can see obvious benefits, since signed Supermasks facilitate a drastic reduction of energy consumption when deploying neural nets and they can contribute to the interpretability, explainability and ultimately trustworthiness of deep learning approaches.

REPRODUCIBILITY STATEMENT

The authors provide the complete code used to conduct all experiments in the supplementary material. An exemplary configuration and requirements file are attached as well, with which all experiments can be easily reproduced. Importantly, all experiments were run on a single GPU, which simplifies reproduction and strengthens the overall benefits of signed Supermasks. The code is written in Python, mainly using tensorflow and numpy. All datasets used in this work are provided by tensorflow here.

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

# A  DERIVATION OF ELUS

In the following paragraphs, we revise the initialization methods of He et al. (2015) and Glorot & Bengio (2010). This is needed because first, we introduce masking and a the ELU activation function. Both concepts alter the assumptions of the cited work. Here, we derive a novel adaption of the common He initialization (He et al., 2015).
In this section, we use the following notation:

- Activation function $\sigma(x)$
- Layer $l$ has $n_l$ neurons. This means that a layer takes $n_{l-1}$ inputs per neuron. In the literature, $n_{l-1}$ is sometimes called *fan-in* and $n_l$ is sometimes referred to as *fan-out*
- $\boldsymbol{W}_l \in \mathbb{R}^{n_l \times n_{l-1}}$, $\boldsymbol{b}_l \in \mathbb{R}^{n_l \times 1}$
- Weighted layer input $\boldsymbol{z}_l = \boldsymbol{W}_l \boldsymbol{o}_{l-1} + \boldsymbol{b}_l$, $\boldsymbol{z}_l \in \mathbb{R}^{n_l \times 1}$
- Layer output $\boldsymbol{o}_l = \sigma(\boldsymbol{z}_l)$, $\boldsymbol{o}_l \in \mathbb{R}^{n_l \times 1}$
- $\mathcal{L}$ denotes the network's loss function
- $\nabla \boldsymbol{o}_l = \frac{\partial \mathcal{L}}{\partial \boldsymbol{o}_l}$
- $\nabla \boldsymbol{z}_l = \frac{\partial \mathcal{L}}{\partial \boldsymbol{z}_l} = \sigma'(\boldsymbol{z}_l) \nabla \boldsymbol{o}_{l+1}$
- In order to maintain clarity, we will denote the transposed weight matrix of layer $l$ $\boldsymbol{W}_l^T$ as $\hat{\boldsymbol{W}}_l$ in the derivation of the backward-pass.
- $\bar{\boldsymbol{W}}_l = \boldsymbol{W}_l \odot \bar{\boldsymbol{M}}_l$. This leads to the suboptimal but necessary choice to denote the transpose of $\bar{\boldsymbol{W}}_l$ as $\tilde{\boldsymbol{W}}_l$ (instead of $\bar{\boldsymbol{W}}_l^T$) to maintain clarity.

We will further assume, that the following holds:

(1) We use the ELU activation function, introduced by Clevert et al. (2015).
(2) Variances of input features $\boldsymbol{x}$ are the same $\mathrm{Var}[x]$.
(3) For each layer $l$, the respective weight matrix $\boldsymbol{W}_l$ is drawn from an independent and symmetric distribution with $\mathbb{E}[\boldsymbol{W}_l] = 0$ and $\mathrm{Var}[\boldsymbol{W}_l]$. It follows that $\mathbb{E}[\boldsymbol{z}_l] = 0$.
(4) For each layer $l$, the respective mask matrix $\boldsymbol{M}_l$ is drawn from an independent and symmetric distribution with $\mathbb{E}[\boldsymbol{M}_l] = 0$. To make the derivations more tangible, we assume $p_0^l = P(\bar{\boldsymbol{M}}_{ij}^l = 0) = 0.5$ and $p_1^l = P(\bar{\boldsymbol{M}}_{ij}^l = 1) = P(\bar{\boldsymbol{M}}_{ij}^l = -1) = 0.25 \ \forall i \in 1, \ldots, n_l, j \in 1, \ldots, n_{l-1}$ at initialization. In general, the variance is $\mathrm{Var}[\bar{\boldsymbol{M}}^l] = 1 - p_0^l$.
(5) For each layer $l$, all elements in $\boldsymbol{z}_l$ and $\boldsymbol{o}_l$ share the same variance $\mathrm{Var}[\boldsymbol{z}_l]$ and $\mathrm{Var}[\boldsymbol{o}_l]$, respectively.

Since we use ELU activation functions in our experiments, the question arises whether we can still use the initialization of He et al. (2015). Clevert et al. (2015) do so and achieve good results. Let us therefore calculate the important values that might alter the assumptions necessary.
Like Glorot & Bengio (2010) and He et al. (2015) differ in the choice of activation function, we can particularly investigate the parts of the derivation where the expected value of the activation function is used. In particular, this is $\mathbb{E}[(\boldsymbol{o}_{l-1})^2]$ in the forward-pass and $\mathbb{E}[(\nabla \boldsymbol{o}_l)^2]$ in the backward-pass. We can calculate $\mathbb{E}[(\boldsymbol{o}_{l-1})^2]$ as follows:

$$\mathbb{E}[(\boldsymbol{o}_{l-1})^2] = \mathbb{E}[(\alpha e^{\boldsymbol{z}_{l-1}} - \alpha)^2] + \mathbb{E}[\boldsymbol{z}_{l-1}^2] \tag{4}$$

From here on, we drop index $l$ for the sake of clarity. For the calculation of $\mathbb{E}[(\alpha e^{\boldsymbol{z}_{l-1}} - \alpha)^2]$, we additionally assume that $\boldsymbol{z}_l \sim \mathcal{N}(0, 1)$, although this is a simplification and is therefore only an approximation. With this, we obtain

$$\begin{aligned}
\mathbb{E}[(\alpha e^{\boldsymbol{z}} - \alpha)^2] &= \int_{-\infty}^{0} (\alpha e^{\boldsymbol{z}} - \alpha)^2 \cdot \frac{1}{\sqrt{2\pi}} e^{-\frac{1}{2}\boldsymbol{z}^2} \, d\boldsymbol{z} \\
&= \frac{\alpha^2}{\sqrt{2\pi}} \int_{-\infty}^{0} (e^{2\boldsymbol{z}} - 2e^{\boldsymbol{z}} + 1) \cdot e^{-\frac{1}{2}\boldsymbol{z}^2} \, d\boldsymbol{z} \\
&= \underbrace{\frac{\alpha^2}{\sqrt{2\pi}} \int_{-\infty}^{0} e^{-\frac{1}{2}\boldsymbol{z}^2 + 2\boldsymbol{z}} \, d\boldsymbol{z}}_{=:(a)} - \underbrace{\frac{2\alpha^2}{\sqrt{2\pi}} \int_{-\infty}^{0} e^{-\frac{1}{2}\boldsymbol{z}^2 + \boldsymbol{z}} \, d\boldsymbol{z}}_{=:(b)} + \underbrace{\frac{\alpha^2}{\sqrt{2\pi}} \int_{-\infty}^{0} e^{-\frac{1}{2}\boldsymbol{z}^2} \, d\boldsymbol{z}}_{=\frac{1}{2}\alpha^2 \mathrm{Var}[\boldsymbol{z}] = \frac{1}{2}\alpha^2}
\end{aligned} \tag{5}$$

We can observe that the last term is half of the area under the standard normal probability density function. Note that it holds that $-ax^2 + bx = -a(x - \frac{b}{2a})^2 + \frac{b^2}{4a}$. With this, we can re-write $(a)$ and $(b)$ as follows

$$(a) = \frac{\alpha^2}{\sqrt{2\pi}} \int_{-\infty}^{0} e^{-\frac{1}{2}(\boldsymbol{z}-2)^2 + 2} \, d\boldsymbol{z} = \frac{\alpha^2 e^2}{\sqrt{2\pi}} \int_{-\infty}^{0} e^{-\frac{1}{2}(\boldsymbol{z}-2)^2} \, d\boldsymbol{z} \tag{6}$$

$$(b) = \frac{2\alpha^2}{\sqrt{2\pi}} \int_{-\infty}^{0} e^{-\frac{1}{2}(\boldsymbol{z}-1)^2 + \frac{1}{2}} \, d\boldsymbol{z} = \frac{2\alpha^2 e^{\frac{1}{2}}}{\sqrt{2\pi}} \int_{-\infty}^{0} e^{-\frac{1}{2}(\boldsymbol{z}-1)^2} \, d\boldsymbol{z} \tag{7}$$

We can see that $(a)$ is the probability density function of $\mathcal{N}(2,1)$ and $(b)$ is the probability density function of $\mathcal{N}(1,1)$, which can be easily evaluated. With $(a) = \alpha^2 e^2 \cdot \Phi(\frac{z-2}{1}) = 0.168102\alpha^2$ and $(b) = 2\alpha^2 e^{\frac{1}{2}} \cdot \Phi(\frac{z-1}{1}) = 0.523157\alpha^2$ we have

$$\mathbb{E}[(\alpha e^{\boldsymbol{z}} - \alpha)^2] = 0.168102\alpha^2 - 0.523157\alpha^2 + 0.5\alpha^2 = 0.144945\alpha^2 := k \tag{8}$$

If we take the case $x > 0$ into account, i.e. $\mathbb{E}[\boldsymbol{z}^2] = \frac{1}{2}\text{Var}[z]$, we have

$$\mathbb{E}[(\boldsymbol{o}_{l-1})^2] = (\frac{1}{2} + k)\text{Var}[\boldsymbol{z}_{l-1}] \tag{9}$$

Let us now move on to calculating $\mathbb{E}[(\nabla \boldsymbol{o}_l)^2]$. Note that the derivative of ELU is simply 1 if $\boldsymbol{z} > 0$ (which is the same as for ReLU) and $\alpha e^{\boldsymbol{z}}$ if $\boldsymbol{z} \leq 0$. Let us look at the squared expectation of the second case:

$$\mathbb{E}[(\sigma'(\boldsymbol{z}))^2] = \mathbb{E}[\alpha^2 e^{2\boldsymbol{z}}] \tag{10}$$

If we assume again, that $\boldsymbol{z}_l \sim \mathcal{N}(0,1)$, we immediately see that we already calculated this expected value in Equation 6. Therefore, we have

$$\mathbb{E}[(\sigma'(\boldsymbol{z}))^2] = 0.168102\alpha^2 := h \tag{11}$$

For the total expected value, we then have

$$\mathbb{E}[(\nabla \boldsymbol{o}_l)^2] = \frac{1}{2} + h \tag{12}$$

To summarize, in comparison to (He et al., 2015), we only have to scale Equations 9 and 12 with the constants $h$ and $k$, respectively. The results above are equal to Equation 15 in He et al. (2015), where they assume a PReLU activation (PReLU as an activation function is also introduced in He et al. (2015)). However, they do not supply a derivation. Therefore we cannot guarantee equality with confidence.

Hence, we are careful with the pronoun "our" (initialization) and we will therefore simply refer to the initialization method that follows from the derivations above as "**ELU**".

Let us now use these results and calculate $\text{Var}[\boldsymbol{W}_l]$.

**Forward-Pass:** We start with the following equation

$$\text{Var}[\boldsymbol{z}_l^j] = \sum_{k=0}^{n_{l-1}} \text{Var}[\bar{\boldsymbol{W}}_l^{jk} \boldsymbol{o}_{l-1}^k] \tag{13}$$

Because $\mathbb{E}[\bar{\boldsymbol{W}}_l] = 0$ we can re-write this as

$$\text{Var}[\boldsymbol{z}_l] = n_{l-1}\text{Var}[\bar{\boldsymbol{W}}_l]\mathbb{E}[(\boldsymbol{o}_{l-1})^2] \tag{14}$$

With Equation 9 it follows that

$$\text{Var}[\boldsymbol{z}_l] = (\frac{1}{2} + k)n_{l-1}\text{Var}[\bar{\boldsymbol{W}}_l]\text{Var}[\boldsymbol{z}_{l-1}] = \text{Var}[\boldsymbol{x}] \prod_{i=1}^{l} (\frac{1}{2} + k)n_{i-1}\text{Var}[\bar{\boldsymbol{W}}_i] \tag{15}$$

If we want to achieve constant variance throughout the network, we can set

$$(\frac{1}{2} + k)n_{l-1}\text{Var}[\boldsymbol{W}_l]\text{Var}[\bar{\boldsymbol{M}}_l] \overset{!}{=} 1$$

$$\Longleftrightarrow \text{Var}[\boldsymbol{W}_l] = \frac{1}{(\frac{1}{2} + k)n_{l-1}(1 - p_0^l)} \tag{16}$$

With $\mathrm{Var}[\bar{M}_l] = p_1^l \cdot (-1)^2 + p_0^l \cdot 0^2 + p_1^l \cdot 1^2 = 0.5$ as stated above and $\alpha = 1$ we have

$$\mathrm{Var}[\boldsymbol{W}_l] = 1.5505 \cdot \frac{2}{n_{l-1}} \tag{17}$$

Note that for normal training, we would not have to account for the variance of the mask, which would result in $\mathrm{Var}[\boldsymbol{W}_l] = 1.5505 \cdot \frac{1}{n_{l-1}}$.

**Backward-Pass:** We know that

$$\mathrm{Var}[\nabla \boldsymbol{o}_l^i] = \sum_{j=0}^{n_l} \mathrm{Var}[\tilde{\boldsymbol{W}}_l^{ij} \underbrace{\sigma'(\boldsymbol{z}_l^j) \nabla \boldsymbol{o}_{l+1}^j}_{=\nabla \boldsymbol{z}_l^j}] \tag{18}$$

Recall that ELU derivative is

$$\sigma'(\boldsymbol{z}_l) = \begin{cases} 1, & \boldsymbol{z}_l \geq 0 \\ \alpha e^{\boldsymbol{z}_l}, & \boldsymbol{z}_l < 0 \end{cases} \tag{19}$$

We know from Equation 12 that $\mathbb{E}[(\sigma'(\boldsymbol{z}_l))^2] = \frac{1}{2} + h$ and $\mathbb{E}[\nabla \boldsymbol{o}_{l+1}] = 0$. It follows, that $\mathrm{Var}[\nabla \boldsymbol{o}_{l+1}] = \mathbb{E}[(\nabla \boldsymbol{o}_{l+1})^2]$, resulting in

$$\mathrm{Var}[\nabla \boldsymbol{z}_l] = (\frac{1}{2} + h)\mathrm{Var}[\nabla \boldsymbol{o}_{l+1}] \tag{20}$$

With the obtained results we can now re-write Equation 18 as

$$\mathrm{Var}[\nabla \boldsymbol{o}_l] = (\frac{1}{2} + h)n_l \mathrm{Var}[\tilde{\boldsymbol{W}}_l]\mathrm{Var}[\nabla \boldsymbol{o}_{l+1}] \tag{21}$$

Herewith we are able to calculate the variance throughout all layers as

$$\mathrm{Var}[\nabla \boldsymbol{o}_1] = \mathrm{Var}[\nabla \boldsymbol{o}_{L+1}] \prod_{l=1}^{L} (\frac{1}{2} + h)n_l \mathrm{Var}[\tilde{\boldsymbol{W}}_l] \tag{22}$$

Like in the forward-pass, an easy way to keep the variance constant is to set

$$(\frac{1}{2} + h)n_l \mathrm{Var}[\tilde{\boldsymbol{W}}_l] \stackrel{!}{=} 1$$
$$\Longleftrightarrow \mathrm{Var}[\boldsymbol{W}_l] = \frac{1}{(\frac{1}{2} + h)n_l(1 - p_0^l)} = 1.49678 \cdot \frac{2}{n_l} \tag{23}$$

As above, if we use this initialization for training a normal neural network, we would use $\mathrm{Var}[\boldsymbol{W}_l] = 1.49678 \odot \frac{1}{n_l}$.

We will follow the intuition of He et al. (2015) by leaving the choice of Equation 23 or 17 to the user, as their initialization scheme has proven to outperform Glorot & Bengio (2010) in practice and takes the CNN-architecture into account. However, since Equations 23 and 17 are relatively close to each other, we suggest to use the following variance for the sake of simplicity

$$\mathrm{Var}[\boldsymbol{W}_l] = \frac{1.5}{n_l(1 - p_0^l)} \tag{24}$$

## B  MODELS AND HYPERPARAMETERS

Table 7: Neural Network architectures used in the experiments of this work. To be able to compare the results to the existing literature, the architectures are equal to Frankle & Carbin (2018); Zhou et al. (2019) in addition to the Conv8 model used in Ramanujan et al. (2020). Each CNN performs various convolutions (*pool* denotes max-pooling) with stated number of filters, followed by fully connected (FC) layers with specified output neurons. For the CNNs, we always use a filter size of $3 \times 3$. Total parameter count for each architecture is reported in the last row.

|  | FCN | Conv2 | Conv4 | Conv6 | Conv8 |
|---|---|---|---|---|---|
| Conv Layers |  | 64, 64, pool | 64, 64, pool
128, 128, pool | 64, 64, pool
128, 128, pool
256, 256, pool | 64, 64, pool
128, 128, pool
256, 256, pool
512, 512, pool |
| FC Layers | 300,100,10 | 256, 256, 10 | 256, 256, 10 | 256, 256, 10 | 256, 256, 10 |
| Parameter Count | 266.200 | 4.300.992 | 2.425.024 | 2.261.184 | 5.275.840 |

Table 8: Hyperparameter choices for training the *baseline* models of FCN and Conv2 - Conv8. For simplicity, we chose parameters that work well on all models

|  | FCN | Conv2 | Conv4 | Conv6 | Conv8 |
|---|---|---|---|---|---|
| Learning Rate | 0.008 | 0.008 | 0.008 | 0.01 | 0.002 |
| Decay Rate / Step | .96/10 | .96/5 | .96/10 | .96/10 | .96/10 |
| Red. LR on Plat. (Patience) | - | - | - | - | - |
| Weight Decay | 7e-4 | 7e-4 | 7e-4 | 7e-4 | 3e-4 |
| Momentum | .9 | .9 | .9 | .9 | .9 |
| Iterations | 50 | 50 | 50 | 50 | 50 |
| Optimizer | SGD | SGD | SGD | SGD | SGD |

Table 9: Hyperparameter choices for training the *baseline* ResNet models.

|  | ResNet20 | ResNet56 | ResNet110 |
|---|---|---|---|
| Learning Rate | 0.2 | 0.15 | 0.15 |
| Decay Rate / Step | - | - | - |
| Red. LR on Plat. (Patience) | .2 (15) | .2 (15) | .2 (15) |
| Weight Decay | 1e-4 | 1e-4 | 1e-4 |
| Momentum | .9 | .9 | .9 |
| Iterations | 80 | 90 | 90 |
| Optimizer | SGD | SGD | SGD |

Table 10: Hyperparameter choices for training the *signed Supermask* models of FCN and Conv2 - Conv8. For simplicity, we chose parameters that work well on all models, except for Conv2 which suffered overfitting and we therefore reduced the learning rate.

|  | ResNet20 | ResNet56 | ResNet110 |
|---|---|---|---|
| Learning Rate | 0.2 | 0.15 | 0.15 |
| Decay Rate / Step | - | - | - |
| Red. LR on Plat. (Patience) | .2 (15) | .2 (15) | .2 (15) |
| Weight Decay | 1e-4 | 1e-4 | 1e-5 |
| Momentum | .9 | .9 | .9 |
| Iterations | 110 | 120 | 120 |
| Optimizer | SGD | SGD | SGD |
| Initial Pruning Rate | 75% | 85% | 85% |

Table 11: Hyperparameter choices for training the *signed Supermask* ResNet models.

|  | FCN | Conv2 | Conv4 | Conv6 | Conv8 | ResNet20 |
|---|---|---|---|---|---|---|
| Learning Rate | 0.05 | 0.02 | 0.05 | 0.05 | 0.05 | 0.2 |
| Decay Rate / Step | .96/10 | .96/5 | .96/10 | .96/10 | .96/10 | - |
| Red. LR on Plat. (Patience) | - | - | - | - | - | .2 (15) |
| Weight Decay | 5e-4 | 5e-4 | 5e-4 | 5e-4 | 5e-4 | 1e-4 |
| Momentum | .9 | .9 | .9 | .9 | .9 | .9 |
| Iterations | 100 | 100 | 100 | 100 | 100 | 110 |
| Optimizer | SGD | SGD | SGD | SGD | SGD | SGD |
| Initial Pruning Rate | 6.3% | 29.5% | 19.3% | 11.9% | 12.9% | 75% |

## C   FURTHER EXPERIMENTAL RESULTS

Table 12: CNN Signed Supermask: Average test accuracy, test loss and required training time per epoch for each model architecture and weight initialization method. The 5% and 95% quantiles are given as well.

| Architecture | Init | Test Acc. [%] | Test Loss | Time/Epoch [s] |
|---|---|---|---|---|
| Conv2 - *Baseline* | He | 68.79 [68.35, 69.19] | 0.9083 [0.8986, 0.9166] | 2.87 [2.86, 2.88] |
| | ELU | 68.60 [68.23, 68.99] | 0.9189 [0.9047, 0.9331] | 2.88 [2.85, 2.97] |
| | Xavier | 67.93 [67.39, 68.41] | 0.9535 [0.9281, 0.9716] | 2.86 [2.84, 2.88] |
| Conv2 - *Si. Su.* | He | 66.58 [65.92, 67.15] | 0.9596 [0.9452, 0.9741] | 3.16 [3.14, 3.18] |
| | ELUS | 68.37 [67.74, 69.04] | 0.9784 [0.9524, 1.0090] | 3.16 [3.14, 3.18] |
| | Xavier | 54.24 [53.36, 55.39] | 1.2862 [1.2608, 1.3067] | 3.20 [3.15, 3.31] |
| Conv4 - *Baseline* | He | 75.03 [74.71, 75.42] | 0.7266 [0.7200, 0.7347] | 3.99 [3.97, 4.01] |
| | ELU | 75.00 [74.65, 75.29] | 0.7281 [0.7213, 0.7363] | 3.98 [3.97, 4.00] |
| | Xavier | 74.20 [73.68, 74.75] | 0.7542 [0.7390, 0.7714] | 3.99 [3.96, 4.01] |
| Conv4 - *Si. Su.* | He | 75.92 [74.93, 76.78] | 0.8084 [0.7695, 0.8438] | 4.22 [4.19, 4.26] |
| | ELUS | 77.40 [76.73, 78.25] | 0.8309 [0.7863, 0.8721] | 4.22 [4.20, 4.25] |
| | Xavier | 73.99 [73.47, 74.64] | 0.7544 [0.7390, 0.7680] | 2.51 [4.21, 4.34] |
| Conv6 - *Baseline* | He | 75.54 [74.89, 76.03] | 0.7159 [0.7001, 0.7324] | 4.98 [4.92, 5.32] |
| | ELU | 75.91 [75.43, 76.36] | 0.7009 [0.6914, 0.7159] | 4.84 [4.82, 4.85] |
| | Xavier | 75.54 [74.98, 76.13] | 0.7151 [0.7007, 0.7300] | 4.94 [4.92, 4.96] |
| Conv6 - *Si. Su.* | He | 78.42 [77.57, 79.27] | 0.6544 [0.6306, 0.6791] | 5.07 [5.05, 5.10] |
| | ELUS | 79.17 [78.05, 80.49] | 0.6685 [0.6277, 0.7151] | 5.14 [5.09, 5.21] |
| | Xavier | 78.47 [77.52, 79.23] | 0.6724 [0.6397, 0.7252] | 5.12 [5.09, 5.20] |
| Conv8 - *Baseline* | He | 73.82 [70.18, 78.29] | 0.6830 [0.6399, 0.7440] | 6.38 [6.27 , 6.89] |
| | ELU | 76.24 [75.14, 77.18] | 0.7787 [0.7267, 0.8374] | 6.27 [6.26 , 6.29] |
| | Xavier | 73.53 [71.82, 74.85] | 0.9133 [0.8317, 1.0212] | 6.31 [6.27 , 6.39] |
| Conv8 - *Si. Su.* | He | 78.92 [78.17, 79.77] | 0.6194 [0.5992, 0.6430] | 6.67 [6.61 , 6.74] |
| | ELUS | 80.91 [79.93, 81.71] | 0.6057 [0.5769, 0.6511] | 6.65 [6.60 , 6.77] |
| | Xavier | 78.51 [76.28, 79.92] | 0.6360 [0.5957, 0.7147] | 6.68 [6.64 , 6.73] |

Table 13: **ResNet Signed Supermask**: Additional training time (abbreviated "TT" below) and compression rate compared to the respective baselines. Signed Supermask require 3 to 14.5% more training time, which is made up for by the higher compression rate. Unsurprisingly, pruning rate correlates highly with the compression rate.

| | Add. TT/Epoch [%] | Compression Rate [%] |
|---|---|---|
| ResNet20 | 3.42 | 57.65 |
| ResNet20x1.5 | 8.53 | 76.08 |
| ResNet20x2.0 | 13.69 | 84.58 |
| ResNet20x2.5 | 14.42 | 89.13 |
| ResNet20x3.0 | 12.57 | 91.85 |

Table 14: Comparison with a broader literature periphery with regard to the total number of remaining weights on CIFAR-10. We show those results that are also compared with other obtained results in the respective papers. Please note that, as already stated in the introduction, a direct comparison of all those methods is not possible for multiple reasons: first, complexity between the final result varies. As an example, pruned networks allow trained weights whereas a BNN only allows for binary weights and activations. Second, many works experiment with different networks and the usage of additional layers such as batch normalization or dropout. If you start with a very overparameterized network, probability theory gives the network much higher chances to include a better subnetwork than a very small one. On top of that, batch normalization or dropout can have a big impact on the predictive performance, which might hide the true performance of the proposed method.

The methods we compare our signed Supermasks to, are: Supermasks Zhou et al. (2019), edge-popup Ramanujan et al. (2020), IteRand Chijiwa et al. (2021), MPT-1/32 Diffenderfer & Kailkhura (2021), FORCE de Jorge et al. (2020), SET Mocanu et al. (2018), RigL Evci et al. (2019), GraNet Liu et al. (2021), TWN Li et al. (2016a), TTQ Zhu et al. (2016), SCA Deng & Zhang (2020), LR-Net Shayer et al. (2017), BinaryConnect Courbariaux et al. (2015), BNN Courbariaux et al. (2016) and BBG Shen et al. (2019). Considered models are Conv6 and Conv8, VGG-19, ResNet-20, ResNet-18 and Wide ResNet 22-2. VGG-Small summarizes all small VGG-like architectures, i.e. they can slightly differ. We refer to the respective work for details. The handling of the weights can be categorized as either a binary mask ("Masked (B)"), pruned weights where the weight values change during training, ternarized and binarized weights as well as ternary masks ("Masked (T)") as is the case for our signed Supermasks.

| Method | Model | Weights | Acc. [%] | Init. Weights | Rem. Weights |
|---|---|---|---|---|---|
| Supermask | Conv6 | Masked (B) | 76.5 | 2.3 M | 0.25 M - 2.1 M |
| edge-popup | Conv8 | Masked (B) | 86 | 5.3 M | 2.6 M |
| IteRand | Conv6 x2 | Masked (B) | 90 | 4.6 M | 2.3 M |
| MPT-1/32 | Conv8 | Masked (B) | 91.48 | 5.3 M | 0.23 M |
| MPT-1/32 + BN | ResNet-18 | Masked (B) | 94.8 | 11.2 M | 2.2 M |
| FORCE | VGG-19 | Masked (B) | 90 | 20 M | 0.1 M |
| SET | VGG-Small | Pruned | 88 | 8.7 M | 0.47 M |
| RigL | Wide ResNet 22-2 | Pruned | 94 | 26.8 M | 13.4 M |
| GraNet | VGG-19 | Pruned | 93.1 | 20 M | 1 M |
| TWN | VGG-Small | Ternarized | 92.56 | 5.1 M | 5.1 M |
| TTQ | ResNet-20 | Ternarized | 91.13 | 0.27 M | 0.27 M |
| SCA | VGG-Small | Ternarized | 93.41 | 4.9 M | 4.9 M |
| LR-Net | VGG-Small | Ternarized | 93.26 | 5.1 M | 5.1 M |
| BinaryConnect | VGG-Small | Binarized | 91.7 | 4.6 M | 4.6 M |
| BNN | VGG-Small | Binarized | 88.6 | 4.6 M | 4.6 M |
| BBG | ResNet-20 | Binarized | 85.3 | 0.27 M | 0.27 M |
| Sig. Supermask | Conv8 | Masked (T) | 80.91 | 5.3 M | 0.061 M |
| Sig. Supermask + BN | ResNet-20 | Masked (T) | 83.38 | 0.27 M | 0.038 M |

Binary Supermask

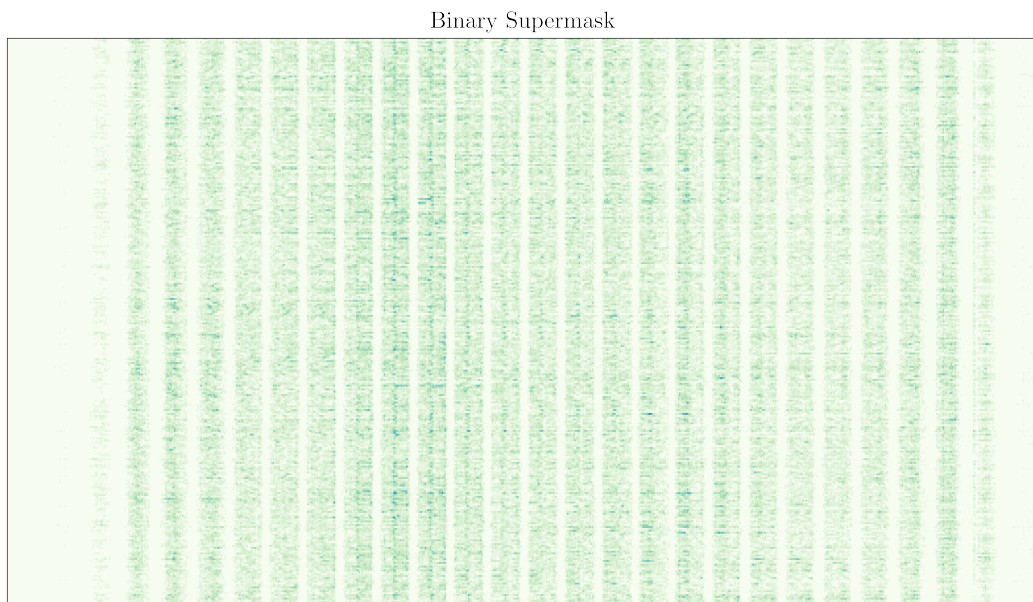

(Absolute) Signed Supermask

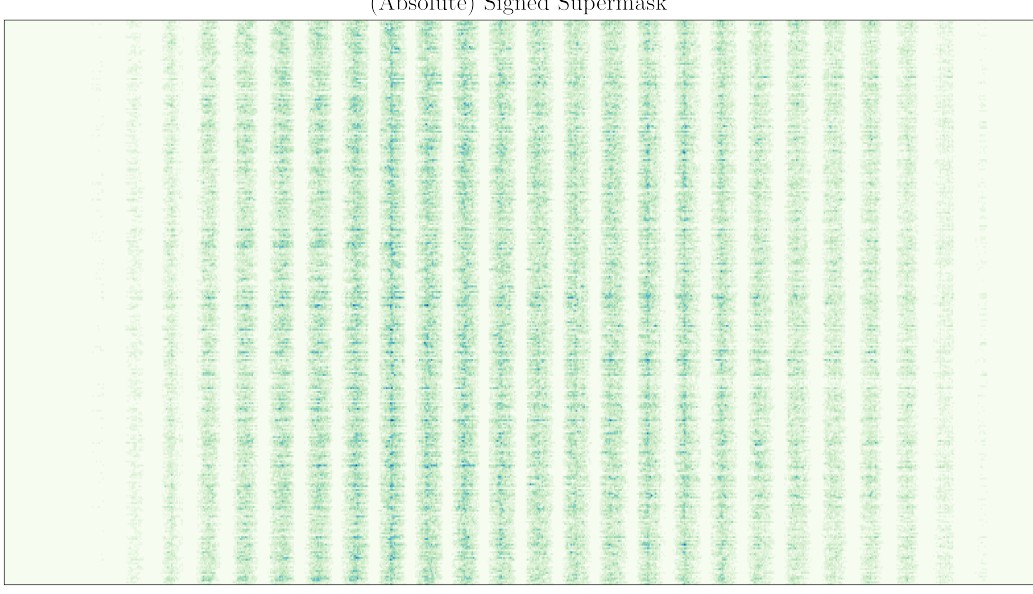

Figure 3: **FCN (Signed) Supermask:** The figure seeks to draw a rough comparison between a FCN trained with a (ELUS) binary Supermask and a (ELUS) signed Supermask. Note that a comparison to Supermasks of Zhou et al. (2019) is not directly feasible as there are other important factors that differ, namely the ELUS initialization, the ELU activation function and the same configuration as used for the ELUS signed Supermask. However, comparing a binary with a signed Supermask can uncover important differences.

Figure 3: The above matrix depicts the sum of 50 trained binary Supermasks, i.e. during the training process the weights could either be pruned or "let through". The same is shown on the bottom for the signed Supermask. A darker shade represents a larger value. In order for any negative and positive masking not to cancel each other out, the absolute masks were summed. Put simply, if a single mask (out of the 50 runs) prunes a weights every time except once, the respective value in the matrices would be 1 and so on. Both versions have an average pruning rate of 97.2% in the first layer. The shown matrices still exhibit a pruning rate of 54.11% in case of the signed Supermask and 43.63% for the binary Supermask, i.e. 54% (43%) of the weights were always pruned. This number is impressive by itself, if we consider that normal neural networks could never reach a similar number. The overall average pruning rates are 3.77% and 4.04% for the signed and binary Supermasks, respectively. While the signed Supermask reached an average performance of 97.48%, the binary Supermask reached an average test accuracy of 97.03%. In other words, the signed Supermask left the same number of weights in the first layer, while the distribution differed more in the other layers but reached a higher performance with roughly 0.3pp less weights. The focus of this examination is to compare the properties of the respective first masks.

At first glance, the binary Supermask looks more erratic, while the signed Supermask inhibits a more ordered structure. The impression is not deceiving: out of the 784 columns (i.e. input neurons), 284 in case of the signed Supermask are completely masked, wheres the binary Supermasks only masks 144 (we strongly assume, that the fact that one number is almost double the other is a coincidence). That is, those columns were masked in every considered mask, which shows the robustness of Supermasks in general. Even if we allow 100 outliers (that is, a column-wise maximum sum of 100) per column, binary Supermasks only count 260 of those columns (364 for signed Supermasks). We can observe this fact in two parts: the signed Supermask is more sparse at the edges and the 0-columns in the center are wider and less noisy. This leads to the conclusion that, although on average having the same pruning rate, signed Supermasks are more consistent (otherwise we would count fewer 0-columns) and more understanding of the given data, as it focuses on specific inputs and mostly disregards the image background, which (in the flattened vector) is positioned at the sides. This strongly suggests that not the pruning rate alone is of importance, but also the distribution of pruning per layer.

To summarize, although a comparison is not possible without compromise, we can see that the binary Supermask, although exhibiting the same average pruning rate is far less consistent compared to the signed Supermask.

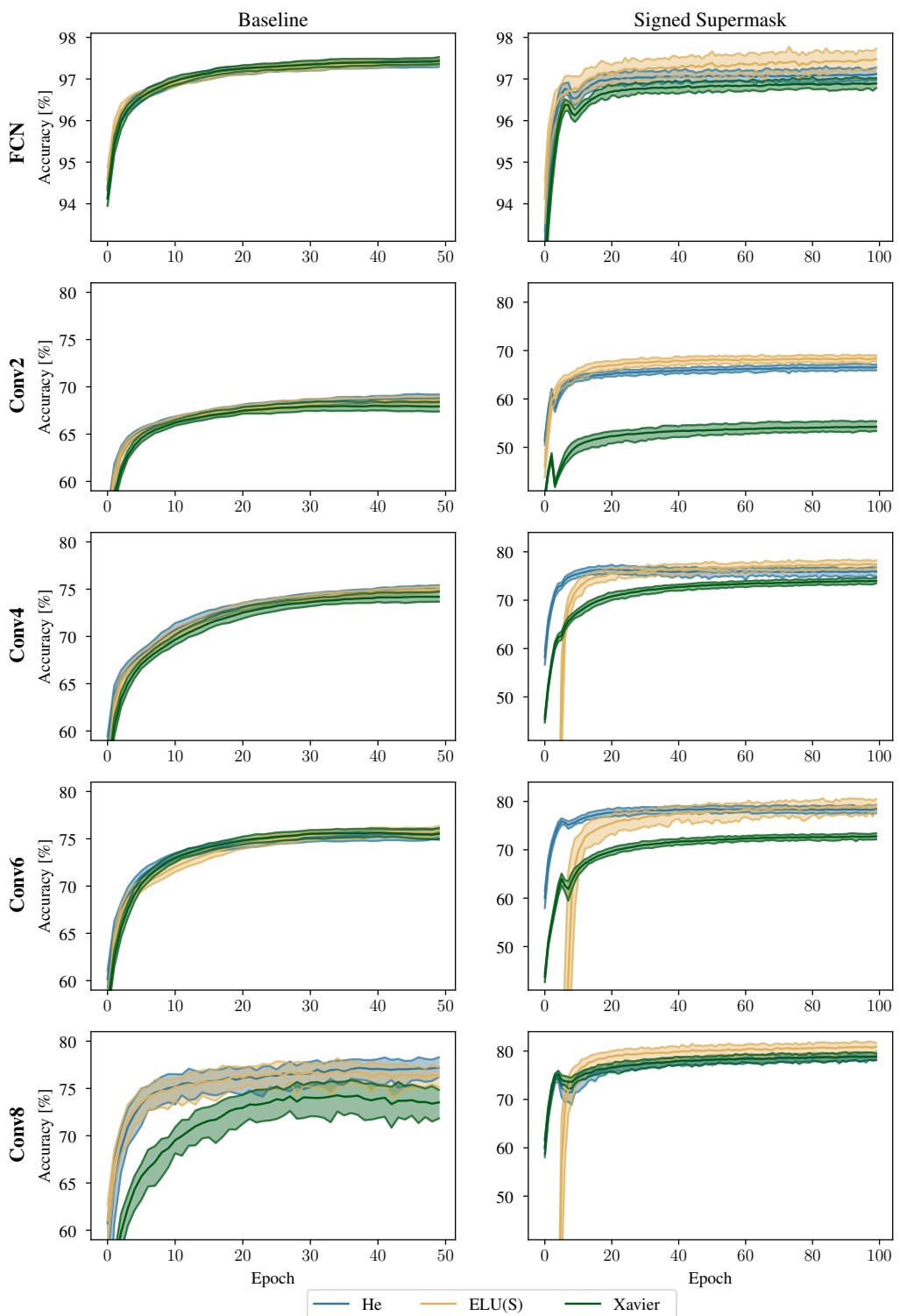

Figure 4: **Signed Supermask: Average test accuracy** over 50 runs of all baseline (left) and signed Supermask (right) models and architectures. We also report the 5% and 95% quantiles. We can observe the larger variance in test accuracy for the signed Supermask models. The ELUS signed Supermask models visually coincides with the baseline or lies above it, whereas He and Xavier cannot reach the same performance. Furthermore, for the deeper models, ELUS needs a few epochs to "warm up".

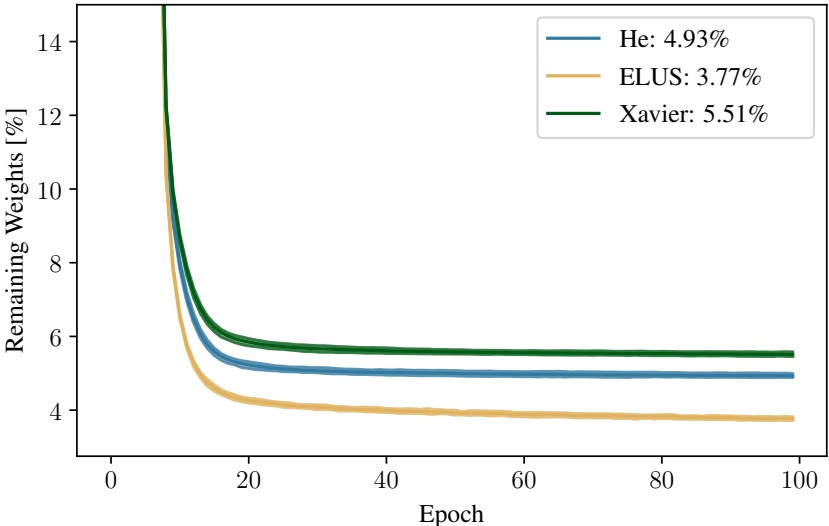

Figure 5: **FCN Signed Supermask: Average ratio of remaining weights.** We also report the 5% and 95% quantile, although the interval is barely visible indicating a robust metric. Weight count drops early in the learning phase and plateaus thereafter.

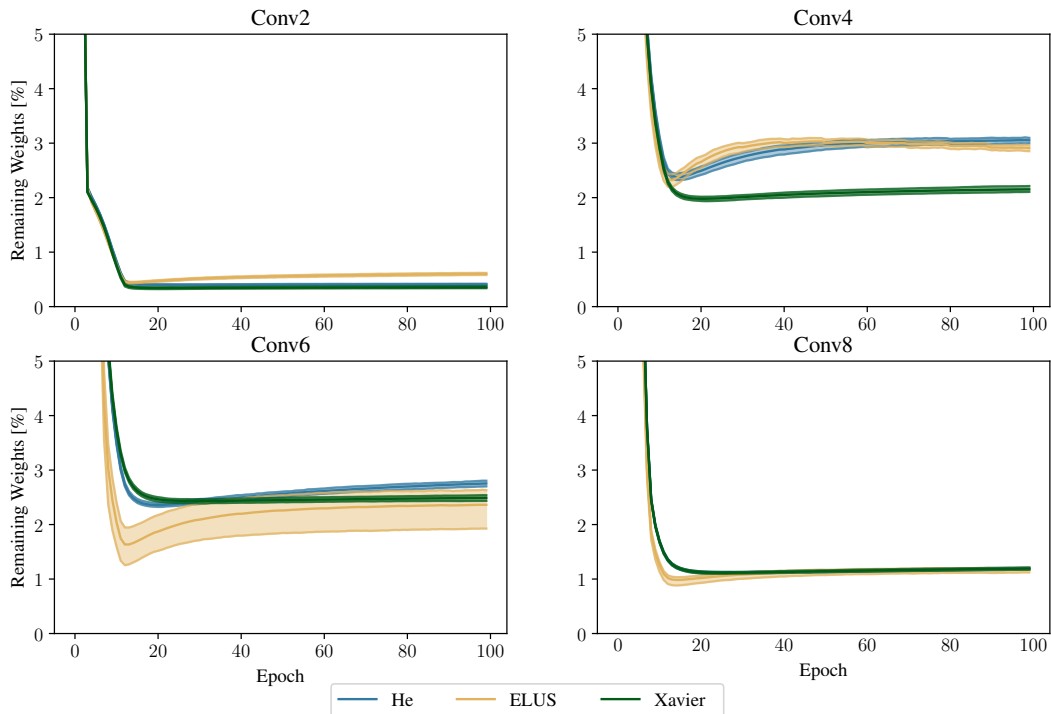

Figure 6: **CNN Signed Supermask: Average ratio of remaining weights** over 50 runs of the CNN signed Supermask models initialized with He, Xavier and ELUS and architectures during training. We also report the 5% and 95% quantiles. Besides for the Conv6 model, all confidence intervals are very narrow. Furthermore, all architectures drop the weights heavily during the first few epochs and plateau thereafter.

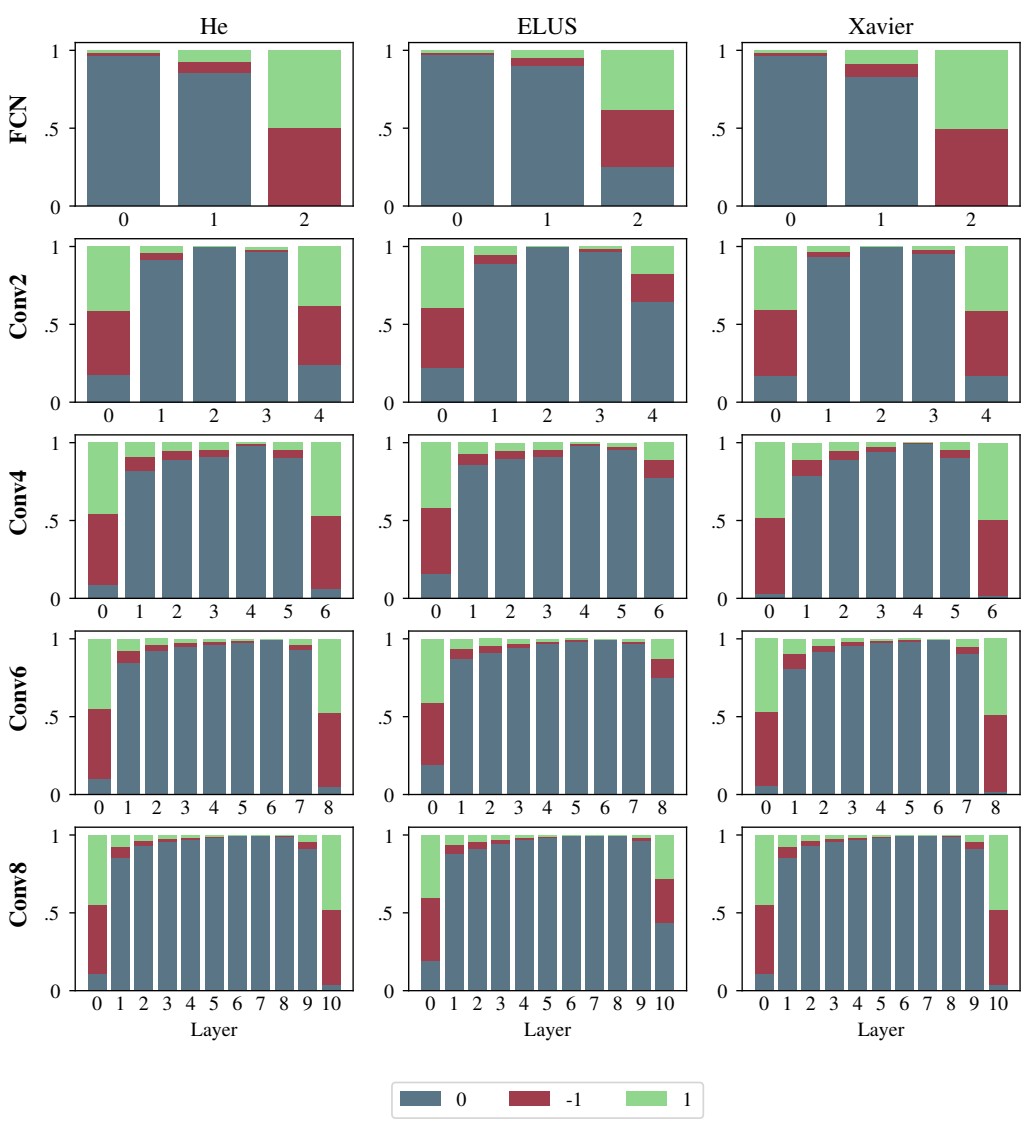

Figure 7: **Signed Supermask: Average mask distribution** over 50 runs of all models and weight initializations. It holds for all configurations that the first layer undergoes only little pruning in contrast to the large hidden layers, where almost all weights are pruned. The behavior in the last layer differs for ELUS: there, the pruning rate is significantly higher in the last layer as compared to He and Xavier. The sparsity correlates with the size of the layer: the larger a layer, the more it is pruned.

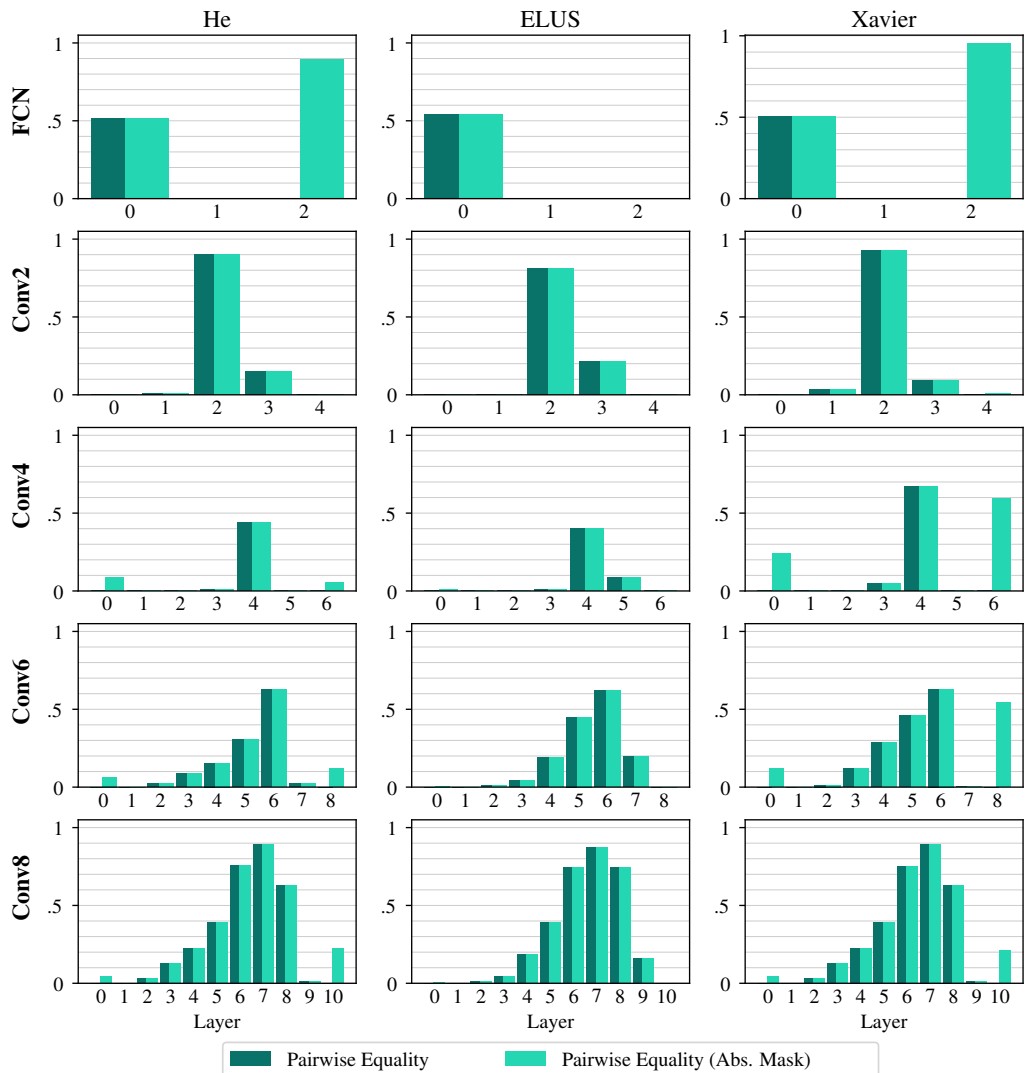

Figure 8: **Signed Supermask: Equality of masks and absolute (i.e. binary) masks** for each architecture and weight initialization over the 50 conducted runs. Only looking at the equality of the signed Supermasks, we can see that the picture is divided into FCN and CNN behavior. For the FCN the first layer is the largest and attains the highest pruning rate (refer to Figure 7), whereas the hidden and output layer are not pruned as much. All initializations produce masks that are similar across all runs in about 50% of all elements. For the CNNs, we have almost no similarity in the first layers which can be partially attributed to the nature of CNNs and the concept of weight-sharing. Here, similarity gradually increases as the layer size and pruning rate are increased, peaking at the respective largest layer.

Taking the absolute signed Supermasks into account, we do not see any meaningful differences in those layers that gain high similarity in the signed Supermask case. However, for the output layers across almost all models that were initialized with He and Xavier we see high similarity. If contrasted with the pruning rate again, its notable that those layers are not pruned much. This can be seen most clearly in the FCNs: He and Xavier only prune the last layer very little, resulting in an absolute signed Supermask that contains many 1's. The same holds for the CNN output layers and some CNN input layers.

These findings suggest that the 0-elements are the main driver of mask similarity, i.e. the same weights are being pruned in every run.

Note that this metric is more takes a more heuristic-approach to analyze the problem. Comparing matrices or tensors is a difficult task. Introducing more precise metrics would be far beyond the scope of this paper.

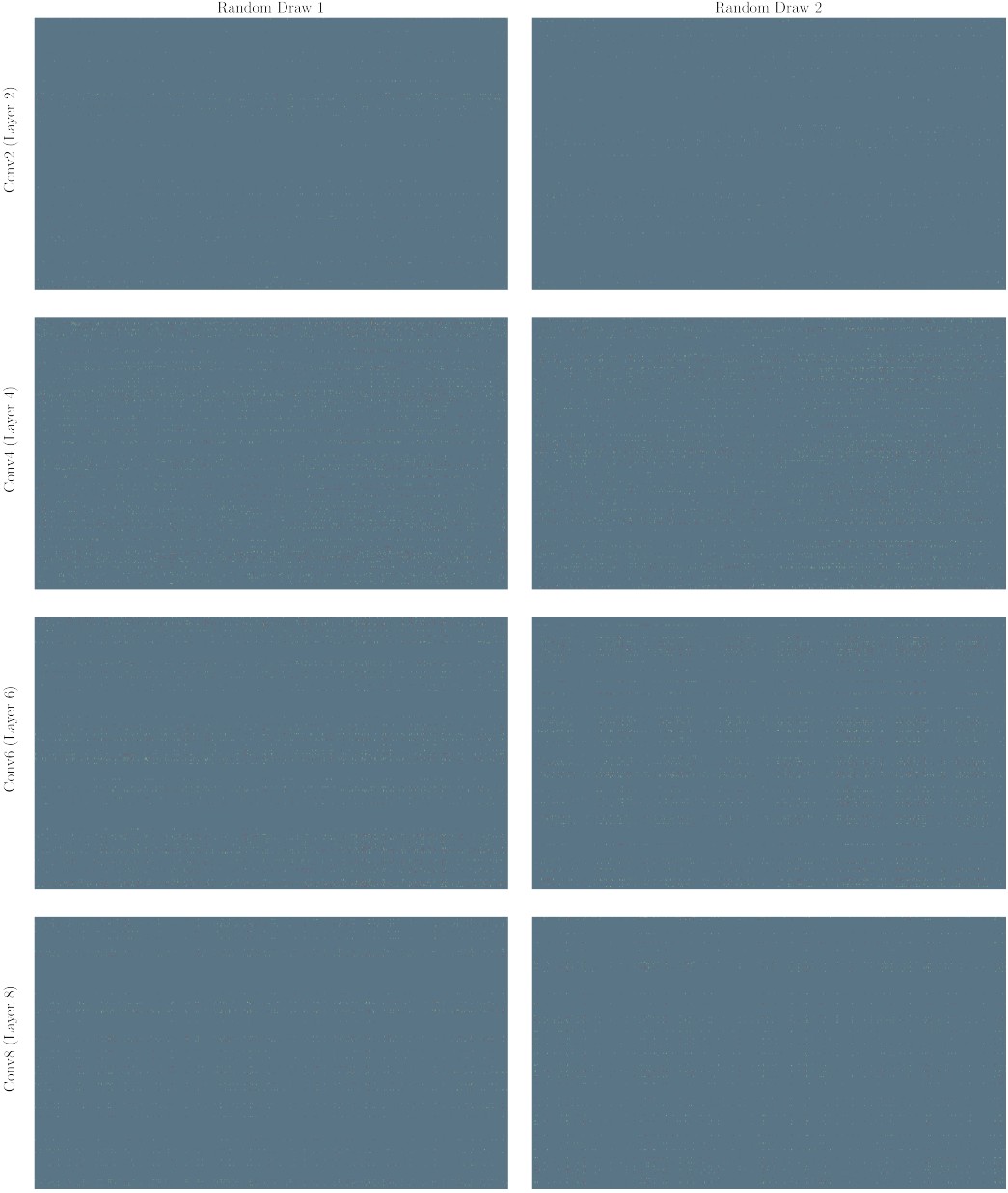

Figure 9: **Conv Signed Supermask:** We show two randomly drawn trained signed Supermasks for each architecture. The chosen layers are the respective first fully connected layer in the respective architecture (which are of different shapes for each architecture). A green dot represents "1", a red dot "-1". For each architecture, we notice unique patterns which are present in both draws. For example, the Conv8 masks show a grid-like pattern which has an area with almost completely pruned weights in the upper part of the mask. The structure is similar for the two Conv6 masks. For Conv4, the masks show more pruning activity on the left part of the mask and a more row-wise structure. Both Conv2 samples are very sparse. The lower third of both masks show very similar patterns, while a wide horizontal band on top is almost completely pruned again.

Interpretation of the masks, e.g. like in Figure 1 is very difficult as the layers here are hidden and depend a lot on the previous and following layers. Still, for each architecture the layers are remarkably similar which cannot be said for standard neural networks.

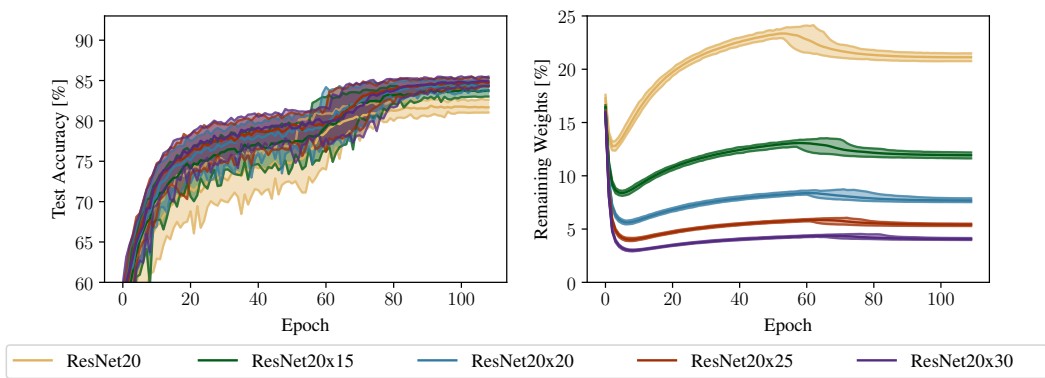

Figure 10: **Signed Supermask: Mean test accuracy and ratio of remaining weights** in addition to the respective 5% and 95% quantiles for ResNet20 and its wider siblings.

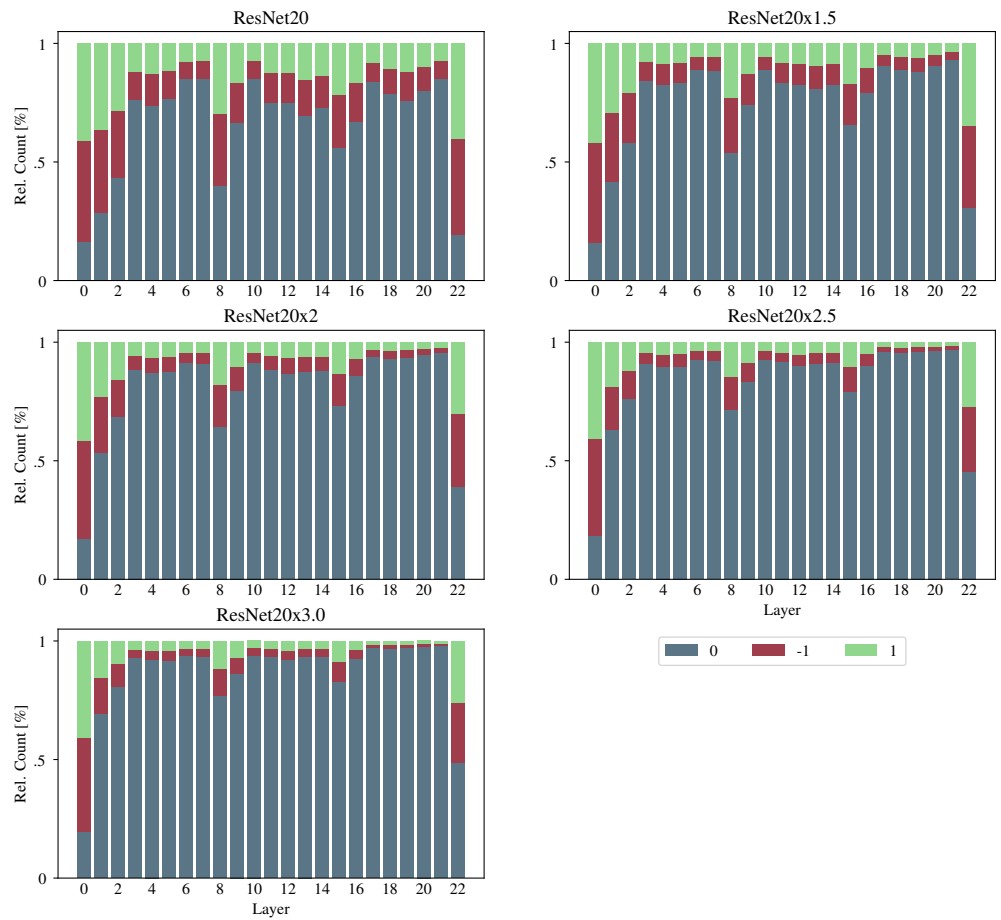

Figure 11: **ResNet Signed Supermask**: Average mask distribution (over the 50 conducted runs) of the investigated ResNet20 models. The wider the layer, the higher the pruning rate. Layers 1, 8 and 15 are 1x1 shortcut layers. Interestingly, those layers are relatively sparse compared to other hidden layers. As with the CNNs, the first and output layer are most sparse.

## D  INFLUENCE OF WEIGHT DISTRIBUTION

In this section we analyze the influence of the weight distribution on overall performance. Therefore, we train additional models with weights drawn from a uniform distributions with the standard thresholds of He (He et al., 2015). As a baseline, we utilize the signed Supermask models with weights as signed constants presented in Section 3.

For both variants, we arbitrarily choose He initialization as we expect similar behavior for the other initialization methods. Hence in this Section, we call the former "He Uniform", the latter "He SC" acting as the baseline. Note that these experiments are not exhaustive, as we only examine the two mentioned distributions. The purpose of this section is to give an intuition on the behavior of different weight distributions. We first analyze the FCN architecture, followed by a summary of the CNN architectures.

**FCN**  Figure 12 displays the average test accuracy and average ratio of remaining weights for both models with the respective 5%- and 95%-quantiles. We can note that the variance of He Uniform is a lot higher than the variance of He SC. Moreover, He Uniform performs on average slightly worse than He SC. The final mean result for He SC is 97.12%, whereas He Uniform achieves 96.77%. A Welch's t-test for significance yields that He SC significantly ($p < 0.01$) outperforms He Uniform. The higher variance of He Uniform is also visible in the ratio of remaining weights. On average, He SC and He Uniform achieve almost the same level of sparsity, however the 5%- and 95%-quantile are almost 2pp apart for He Uniform.

We postulate that a uniform distribution does not have additional value compared to signed constants for a dense neural network in the context of signed Supermask. Apparently, initializing weights with small values deteriorates performance. Recall that neither He SC nor He Uniform are scaled for the use of a signed Supermask. We further speculate that the higher variance is a cause of random initialization such that training cannot nullify the impact of randomness and ill-initialized weights. This leads to the conclusion that the value of a weight is important if not initialized well, as He Uniform holds more weight values compared to He SC and performs worse.

To summarize, using signed constants instead of a uniform distribution provides a much higher level of robustness for the FCN. Our results suggest that for signed Supermask and dense neural networks, a signed constant initialization surpasses the performance of a uniform distribution.

**CNN**  We now investigate the influence of weight distribution on our CNN architectures. Figure 13 visualizes the average test accuracy with the respective 5% and 95% quantiles during training for each model. We first note that He Uniform is not behaving consistently. While with Conv2 and Conv8 the variance of He Uniform is much higher compared to He SC, it is roughly equal for Conv4 and Conv6. The average test accuracy of He Uniform is of the same magnitude as He SC, and we can only register a significantly ($p < 0.01$) better performance for Conv2 He SC compared to He Uniform. This is not sufficient to conclude on one distributions significance for the general case.

The results indicate that the additional weight values delivered by a uniform distribution do not improve the learning capability of a signed Supermask model. This is in line with the findings of the FCN architecture. It seems sufficient to initialize the weight matrix with a well-chosen constant. We further speculate that due to the higher parameter count of the investigated CNN architectures compared to the FCN, the optimizer is to some extent able to neglect the influence of poorly initialized weights.

Let us consider the average ratio of remaining weights over the course of training for each CNN model. Figure 14 visualizes this metric for each network in addition to the 5% and 95% quantiles. Unlike observed in the case of the FCN, we cannot report a high variance for He Uniform within CNN models. In fact, the weight distribution does not seem to have an impact on the level of sparsity, as the ratios move almost equally over the course of training for each model.

Considering both the performance and level of sparsity, we do not find that the choice between uniform and signed constant distribution impacts the end result on average. However, since the test accuracy of He Uniform varies a lot more compared to He SC, the findings suggest to prefer signed constants over a uniform distribution, as signed constants bring positive side-effects such as efficient

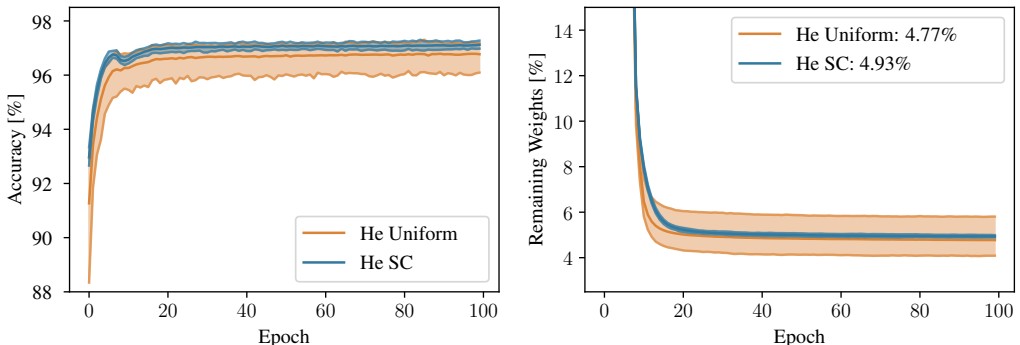

Figure 12: **FCN Signed Supermask: Average test accuracy and average remaining weights** with their respective 5%- and 95%-quantile of a FCN with weights drawn from a uniform distribution and as signed constants ("SC"). We can see that the uniform distribution yields in a much higher variance of both, test accuracy and ratio of remaining weights. Furthermore, drawing weights from a uniform distribution does not improve performance nor sparsity.

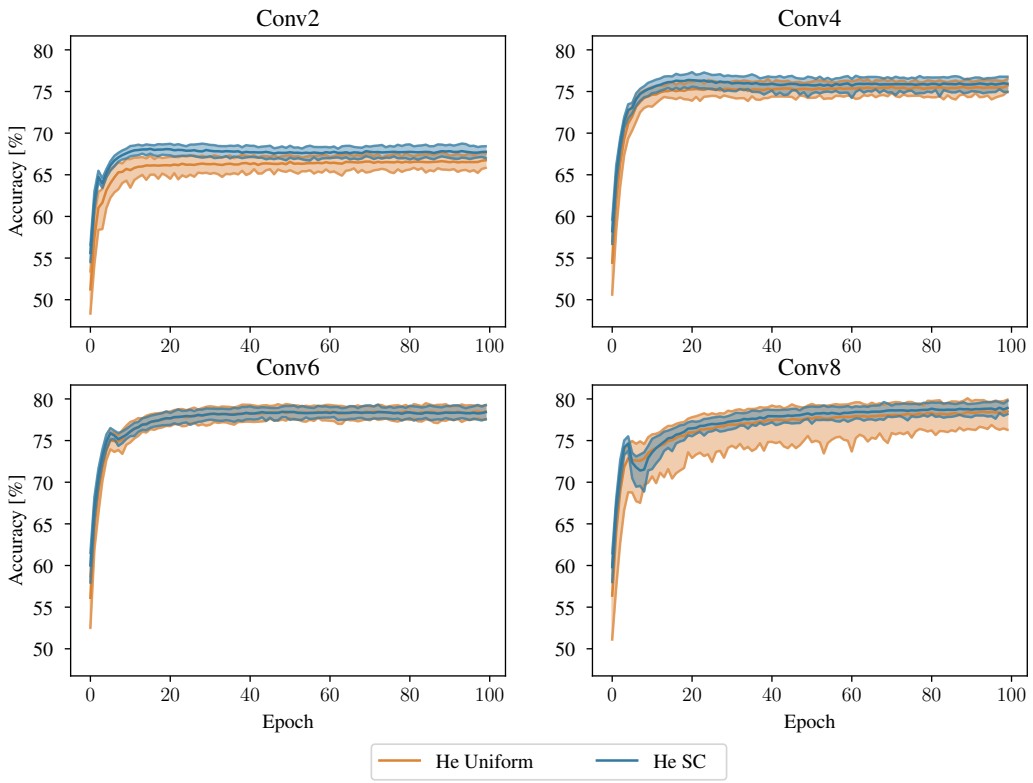

Figure 13: **CNN Signed Supermask: Average test accuracy** for He Uniform and He SC during training. We also report the respective 5%- and 95%-quantile. We find that the variance of He Uniform is at least as large as the variance of He SC, but especially in models Conv2 and Conv8 He Uniform has a much larger variance. He SC outperforms He Uniform in all architectures.

storing and more robust behavior.

To summarize both FCN and CNN behavior, it can be stated that a uniform distribution is not advantageous over a signed constant approach. We further want to re-emphasize the simplicity, yet good performance of signed constants. They have shown a very robust behavior with regards to the

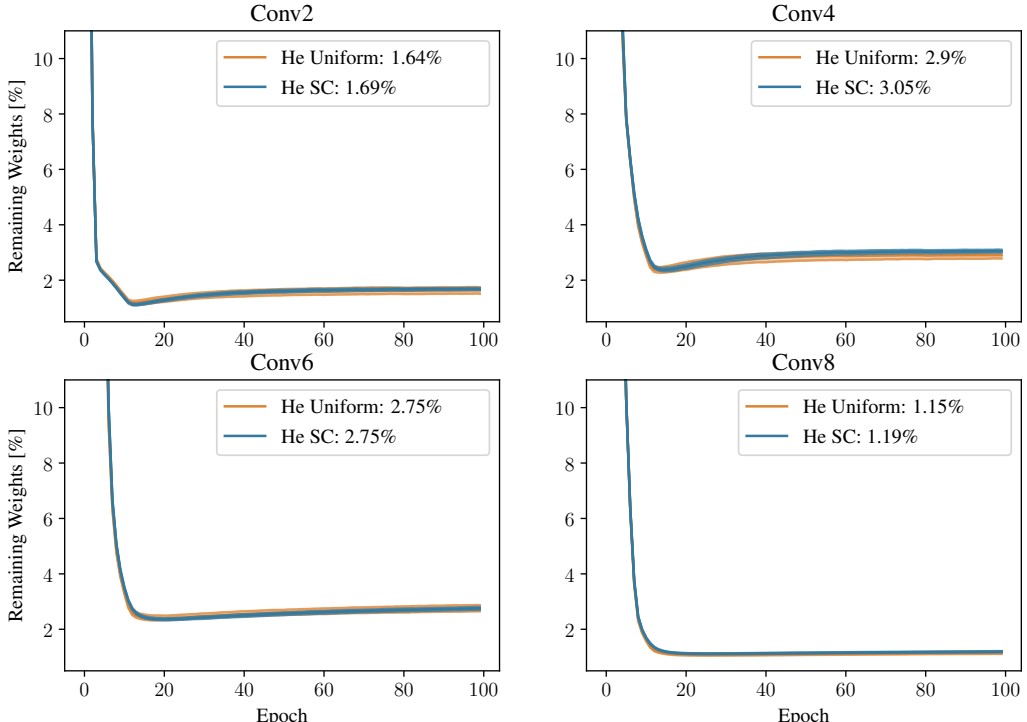

Figure 14: **CNN Signed Supermask: Average ratio of remaining weights** for He Uniform and He SC during training in addition to the respective 5%- and 95%-quantile. We find that the variance of He Uniform is at least as large as the variance of He SC but especially in models Conv2 and Conv8 He Uniform has a much larger variance. He SC outperforms He Uniform in all architectures.

variance in test accuracy while performing equal or better compared to a uniform distribution. It further simplifies the storage of a network as we only have to store each weight's sign, zero and a single weight value for each layer.

## E    INFLUENCE OF MASK INITIALIZATION

We now pursue the question to what extent mask initialization plays a role in performance. Therefore, we use the ELUS models of Sections 3 as a baseline with Xavier uniform mask initialization (abbreviated in this section as "ELUS/Xavier"). Furthermore, we train the same models with ELUS uniform mask initialization (abbreviated in this section as "ELUS/ELUS"). We compare the results separately for the FCN and CNN architectures. As in the last section, we note that these experiments are not exhaustive but give an intuition on the relationship between mask and weight initialization.

**FCN**    Figure 15 visualizes the average test accuracy and average remaining weight ratio with the respective 5% and 95% quantiles. We can note that there is almost no difference in variance and average test accuracy for ELUS/ELUS and ELUS/Xavier. The former achieves an average accuracy of 97.36%, whereas the latter marginally but significantly ($p < 0.01$) exceeds ELUS/ELUS with an average accuracy of 97.48%. As stated in Section 2.1, we presume that mask initialization is not as impactful, as long as weight and mask values are of the same magnitude. The FCN results indicate to support this thesis.
We find a slight deviation in training behavior in the ratio of remaining weights during training. ELUS/ELUS reduces its weight count slightly later in training as ELUS/Xavier does. However, the results are again very alike regarding mean and variance.
We can conclude that for the FCN, the difference in mask initialization did not influence training behavior and robustness meaningfully, yet ELUS/Xavier outperformed ELUS/ELUS slightly but

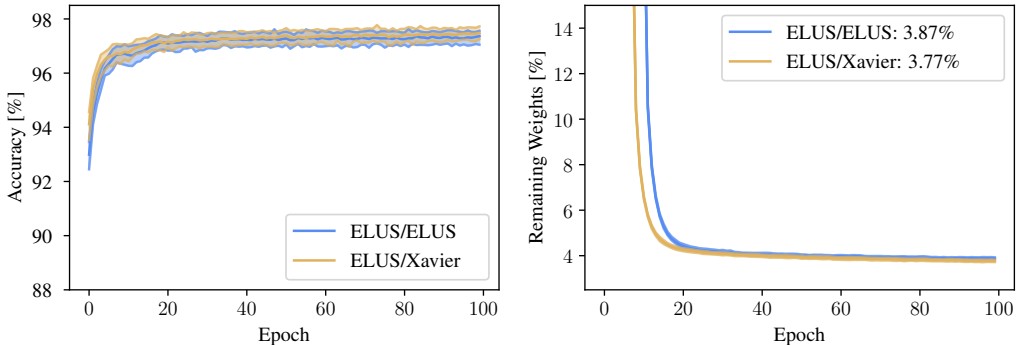

Figure 15: **FCN Signed Supermask: Average test accuracy and average remaining weights** with their respective 5%- and 95%-quantile of a ELUS/ELUS and ELUS/Xavier FCN. We can see that both models behave very similar with regards to performance and variance. ELUS/ELUS drops its weights a few epochs later but the general behavior is equal.

significantly by 0.12pp. Moreover, the average ratio of remaining weights differs only by 0.1pp. Thus, we argue that both weight/mask initialization combinations might be useful, depending on the task at hand.

**CNN**    Our focus shifts towards the CNN architectures. Figure 16 displays the average test accuracy during training with the respective 5% and 95% quantiles for each architecture. We can note a very robust training behavior for both mask initializations, as variance is equal across all models. Observe that only Conv6 ELUS/ELUS performs significantly ($p < 0.01$) better than ELUS/Xavier. We further find that for Conv4, Conv6 and Conv8 ELUS/ELUS begins to optimize effectively a few epochs later than ELUS/Xavier. Yet, this does not influence the final result. We assume this is caused by the optimizer and straight-through estimation, as the gradient might not be approximated properly in the very early learning phase.

Figure 17 visualizes the ratio of remaining weights for each CNN architecture with the respective 5% and 95% quantiles. First, we note that each model has a distinct behavior. Whereas Conv2 ascends its weights rapidly, it then almost linearly reduces the weights further until it plateaus. The other architectures exhibit a similar behaviour towards the end of training, but without the linear section. This is in line with the findings in Section 3. Additionally, we observe that the choice of weight/mask initialization has little influence on the respective curves for each architecture. As already seen with the FCN model, ELUS/ELUS drops its weights a few epochs later than ELUS/Xavier. Moreover, ELUS/Xavier utilizes 0.02pp to 0.4pp weights less than ELUS/ELUS, depending on the network architecture.

Both patterns are consistent for FCN and CNN models. We hypothesize, that the later rise of accuracy and later drop of weight count for ELUS/ELUS is a cause of the gradient estimation. The shift between the two initializations is consistent throughout all architectures. It is furthermore consistent that ELUS/ELUS results in a marginally larger ratio of remaining weights.

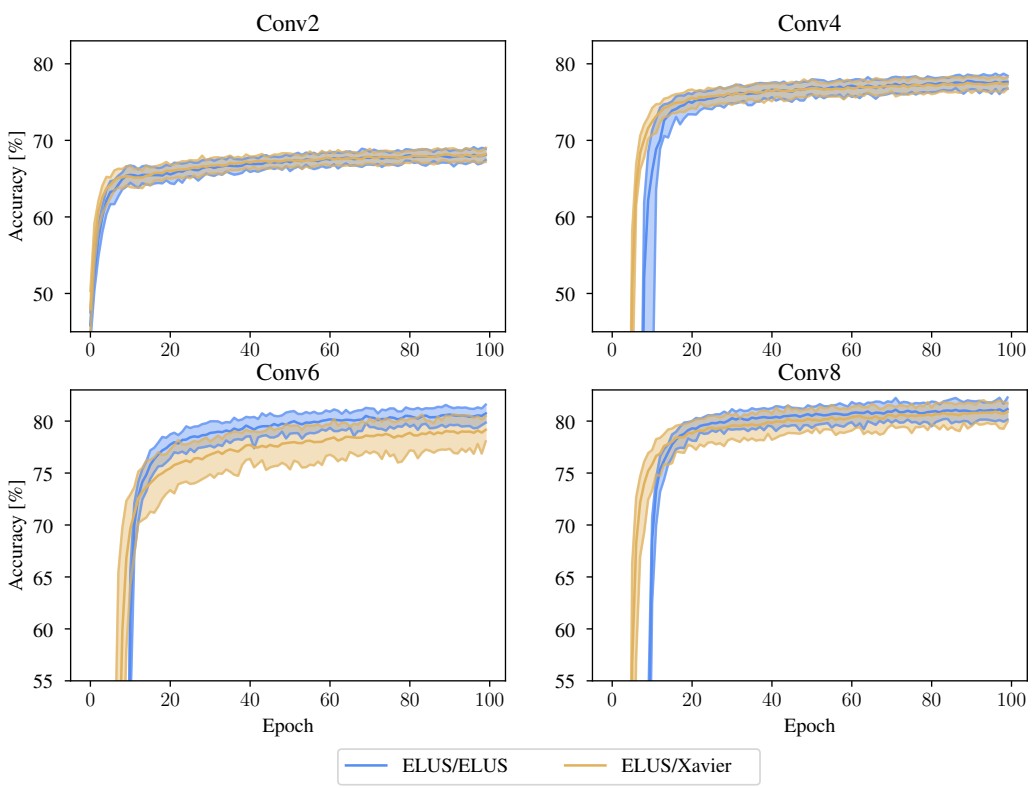

Figure 16: **CNN Signed Supermask: Average test accuracy** for ELUS/ELUS and ELUS/Xavier during training. We also report the respective 5%- and 95%-quantile. We find that for all architectures, both combinations behave and perform very similar. However, ELUS/ELUS visibly exceeds ELUS/Xavier for Conv6.

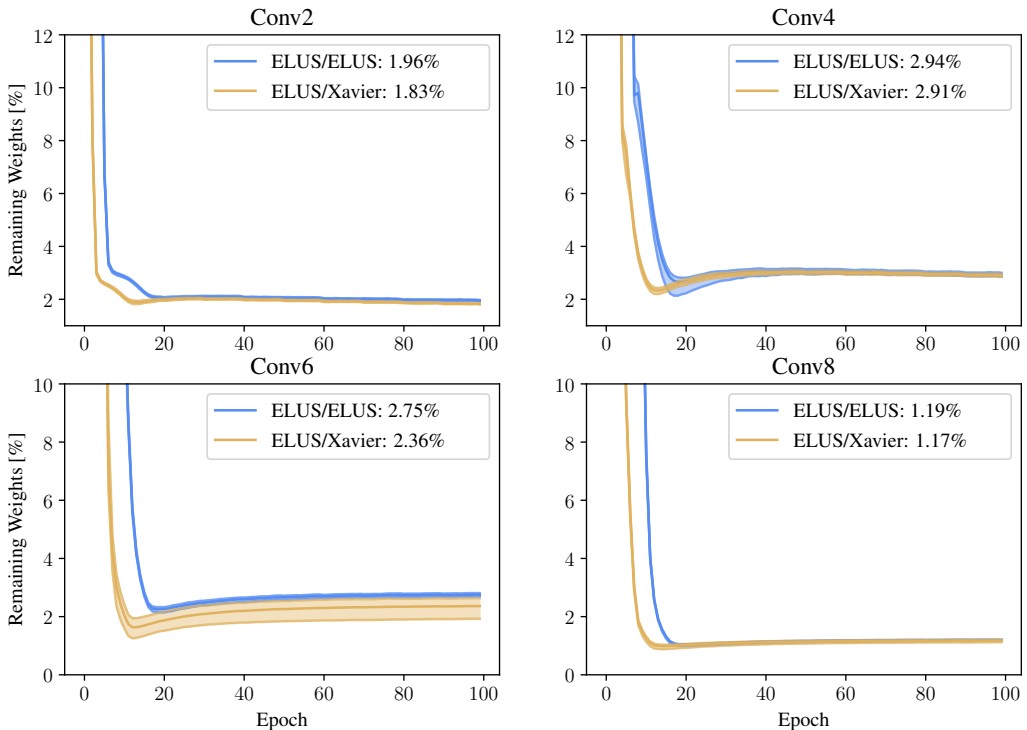

Figure 17: **CNN Signed Supermask: Average ratio of remaining weights** for ELUS/ELUS and ELUS/Xavier during training in addition to the respective 5%- and 95%-quantile. For each architecture, it is visible that the training behavior of both combinations is very similar. Moreover, we can note the different shapes of ascend of the four architectures. ELUS/ELUS drops its weights a few epochs later and achieves a marginally higher ratio at the end of training. Both initialization combinations behave very robust.

## F    How far can we go?

Two important questions still need to be answered. Can we control the pruning rate? And if so, how small can the subnetworks get before the performance is impacted negatively? As we saw in Table 2, Conv2 achieved the lowest pruning rate of all CNN models. However, as described in the beginning of this section, we lowered the learning rate of Conv2 to tackle its tendencies to overfit. Hence, we can investigate if all CNN models show a similar behavior when altering the learning

|  | SiNN 1 | SiNN 2 |
|---|---|---|
| Learning Rate | 0.01 | 0.008 |
| Weight Decay | 5e-4 | 3e-4 |

Table 15: Simple-minded Neural Networks: Altered Hyperparameter choices to minimize weight count during training.

parameters in the same way. Table 15 depicts two altered parameter settings. Why the learning rate somewhat influences pruning is for future work to investigate. In the following, we will call the networks initialized with ELUS and trained with those hyperparameters SiNN (**S**imple-m**i**nded **N**eural **N**etworks) models. As Figure 19 reports, we see that both models prune more than 99% of the original weights, SiNN 1 reaching even pruning rates above 99.5%. Considering the predictive performance we can note, that the performance of SiNN 1 trails significantly behind SiNN 2 for each model.

Let us compare the SiNN models to the baseline models. Figure 18 depicts the test accuracy of the SiNN 2 and baseline models as well as remaining weights of SiNN 2 and ELUS models on the right side. As SiNN 2 achieves higher accuracy than SiNN 1 with only few more weights, we do not further consider SiNN 1 at this point. As the compression rate after training highly correlates with the ratio of remaining weights, we do not show this metric for the sake of brevity. It can be observed that the SiNN 2 models achieve a similar test accuracy with a maximum of about 0.6% of the original weights. This is a further 50% reduction of numbers of weights compared to the ELUS models. Additionally, it is interesting to see, that the remaining absolute weight count of each CNN model is around 14.000 weights. This suggests that this is the absolute minimum of weights that are needed to map the dataset accurately, independently of the model architecture. The performance difference for SiNN 2 between the different architectures then stems from a deeper arrangement of weights, supporting the hypothesis that deeper architectures (with fewer weights) are beneficial. These results reveal three findings: First, we are able to indirectly control the pruning rate of signed Supermask models by adapting the learning parameters. Second, the SiNN 2 models perform similar to their respective baselines. The uncovered subnetworks however, utilize only 0.3% to 0.6% of the original weights. Third, although performance suffers compared to the ELUS models, the signed Supermasks still perform well, even when pushed to the extreme. Ultimately, this demonstrate not only the performance quality of the models, but also how few parameters are actually needed in general to solve a given task well.

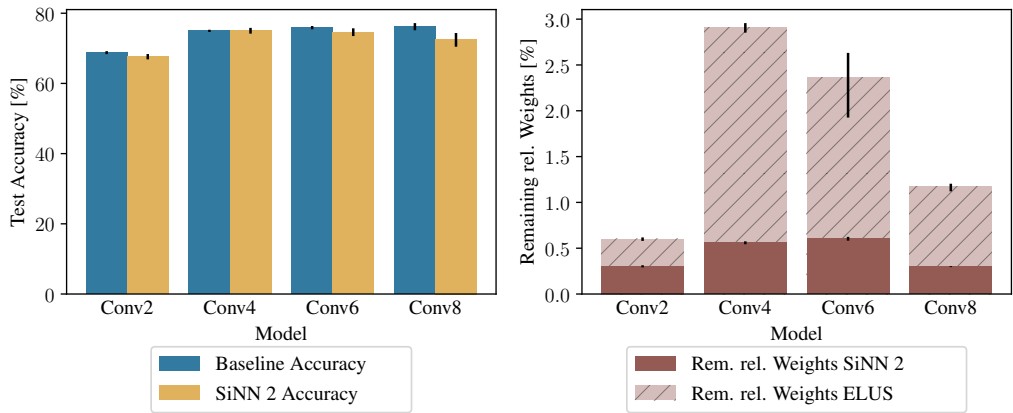

Figure 18: Conv SiNN 2: Test Accuracy (left) for baselines and SiNN 2 models as well as average ratio of remaining weights on the right side for SiNN 2 and ELUS models. 5% and 95% quantiles are reported in the form of error bars for both metrics.

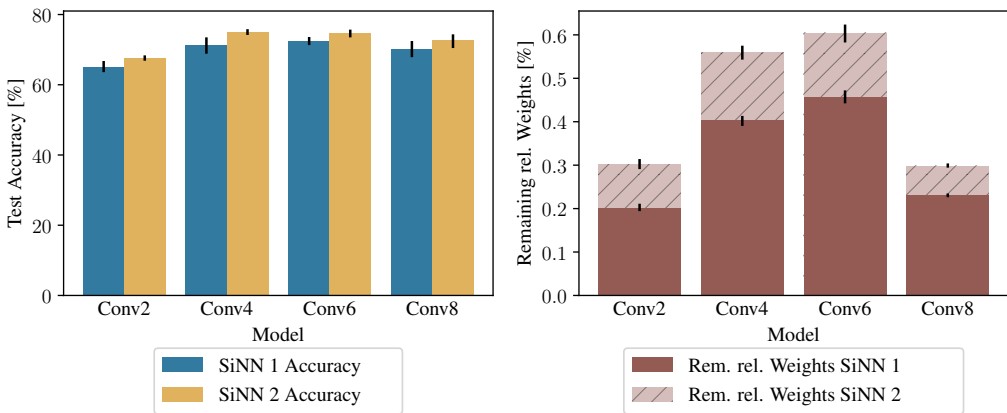

Figure 19: **CNN SiNN Signed Supermasks**: Average SiNN 1 & 2 accuracy (left) and remaining weights (right) over 50 runs of all CNN architectures with error bars indicating the 5% and 95% quantile. SiNN 1 is trumped by SiNN 1 in each architecture in terms of test accuracy. Looking at the right side of the plot, the difference in performance can intuitively be explained by the fewer weights that are left active: for this level of sparsity, a certain amount of weights is clearly needed to yield in acceptable performance. Combining this with the information seen in Figure 21, we can hypothesize that the loss in performance of SiNN 1 compared to SiNN 1 comes from pruning the first layer even more. This further supports the thesis that the first layer in CNNs is crucial for the network's performance, as already stated for the ELUS models. By comparing the masks of SiNN1 and SiNN 2 in greater detail, future work could find e.g. specific weights that influence the performance more than other weights and subsequently draw conclusions on network architectures in general.

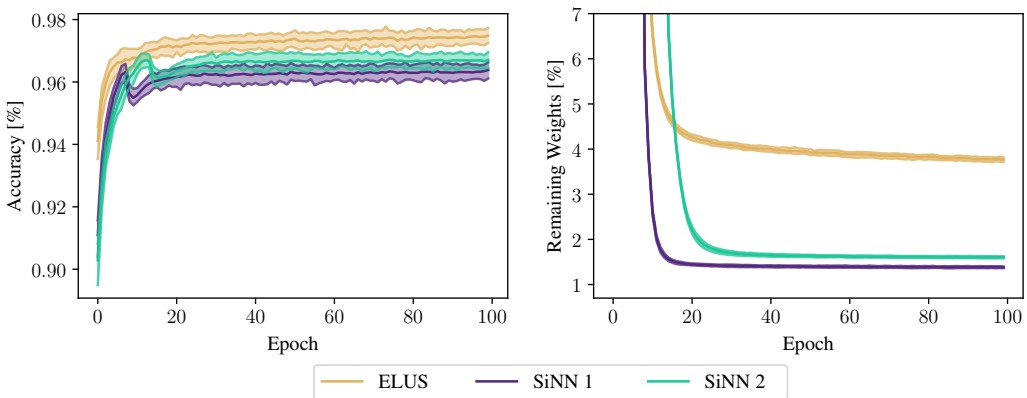

Figure 20: **FCN SiNN Signed Supermask: Average SiNN 1 & 2 accuracy (left) and relative remaining weights (right)** over 50 runs for the FCN model. As a comparison, ELUS is visualized as well. Furthermore, we report the respective 5% and 95% quantiles. SiNN 1 does not reach the test accuracy of neither SiNN 2 nor ELUS, but it also utilizes the fewest weights, around 1.4%. We argue however, that the trade-off of SiNN 2 is better, as the "cost" of performance is only a marginally lower pruning rate.

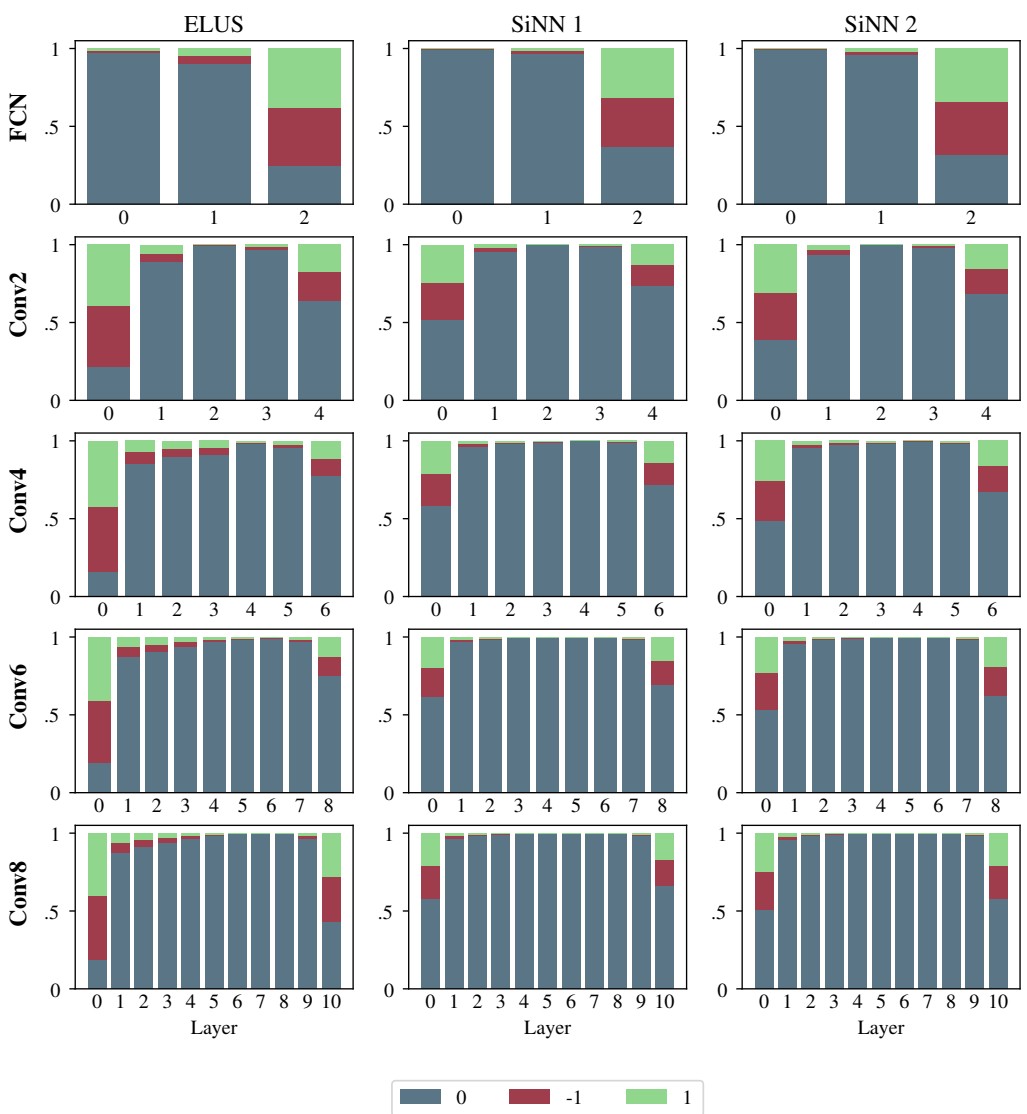

Figure 21: **SiNN Signed Supermask**: Average SiNN 1 & 2 mask distribution over 50 runs of all CNN architectures and ELUS for comparison. For the FCN, the overall picture does not change: SiNN 1 & 2 prune the first two layers even more, while the last layer only experiences little more sparsity. The same is partially true for the CNN architectures: while the last last layer remains on an equal pruning level compared to ELUS, the amount remaining weights in the first layer is considerably smaller. On top, all hidden layers are pruned very heavily. Apart from more aggressive pruning in the first layer for SiNN 1, a difference between SiNN 1 and SiNN 2 cannot be detected visually.

