# OpenReview forum: "Signing the Supermask: Keep, Hide, Invert"
_ICLR.cc/2022/Conference — ICLR 2022 Poster_

### Official Review · Reviewer_okPF · 2021-10-27

**Correctness:** 3
**Technical Novelty And Significance:** 2
**Empirical Novelty And Significance:** 3
**Recommendation:** 5
**Confidence:** 4

**Main Review:**

##########################################################################

Pros:



1. The paper proposes a method to find untrained subnetworks that can approach the performance achieved by trained networks. The performance improvement achieved by allowing weights to flip is interesting.



2. Overall, the paper is well written. Readers can directly grasp the main idea of the paper.



##########################################################################

Cons:

1. My main concern is the motivation and the usage of the proposed method.  As the authors said in the introduction "However, due to their sheer size, the networks not only became difficult to interpret but also problematic to train and use in real-world applications...", I expect a method that can either reduce the number of trained parameters or require a fewer number of FLOPs to be proposed. While the authors claim that the subnetworks discovered are untrained, I find that it requires training mask matrices with the same size as the model weights. Considering that straight-through estimator is used to estimate gradients, I suppose the backward pass is also dense and thus leads to no acceleration for training. Compared with directly training a dense network, I suppose the overall training FLOPs required by Signed Supermask is similar. The larger "TT / Epoch" over the baseline in Table 1 confirms my concern. I believe it is necessary to explain the benefits of Signed Supermask compared with training a dense network and prune or directly training a sparse network. With the sparsity learned by Signed Supermask, I believe even directly training a sparse neural network with static or dynamic sparsity (e.g., Mocanu et al [1], Evci et al [2], Liu et al [3]) can have similar performance, while with much fewer training FLOPs. I encourage the authors to clarify this.

2. There are some related works that are missing in the paper, e.g., Diffenderfer & Kailkhura [4] (accepted by ICLR 2021) and Chijiwa [5] (accepted by NeurIPS 2021). Since the number of works on this topic is quite small, I expect a good submission should at least do a good related work job by introducing them. What's more, as they are aiming to search for untrained subnetworks without adding another dimension, comparisons with these works are encouraged. The current experiments only include comparisons with Zhou2019 and Ramanujan2019 with small convolutional networks, which is too small to draw a solid conclusion. For instance,  Ramanujan2019, [4] and [5] all provide results on large scale dataset ImageNet.

3. As I understand, the performance improvement of Signed Supermask comes from (1) allowing weight flip, (2) ELUS initialization. I expect ELUS  is a universal method that can improve all the related works. However, I didn't see any ablation study of these two components. It is not clear to me if the improvement is caused by one of them or both of them.

4. As mentioned by the authors, the two threshold hyperparameters in equation (1) control the sparsity level. I expect to see more experiments with how these thresholds influence the sparsity level and the corresponding performance of Signed Supermask.

Minor typos:

Instead of starting with a new paragraph, there is an inexplicable blank after some lines. E.g., 2rd paragraph on page 1; 1st paragraph on page 7.

A space is missing before this sentence "The gradient is estimated" on page 3.

Reference:

[1] Mocanu, Decebal Constantin, et al. "Scalable training of artificial neural networks with adaptive sparse connectivity inspired by network science." Nature communications 9.1 (2018): 1-12.

[2] Evci, Utku, et al. "Rigging the lottery: Making all tickets winners." International Conference on Machine Learning. PMLR, 2020.

[3] Shiwei  Liu, et al. ''Sparse training via boost-ing pruning plasticity with neuroregeneration.'' NeurIPS, 2021.

[4] Diffenderfer, J., & Kailkhura, B. (2021). Multi-prize lottery ticket hypothesis: Finding accurate binary neural networks by pruning a randomly weighted network. ICLR, 2021.

[5] Chijiwa, Daiki, et al. "Pruning Randomly Initialized Neural Networks with Iterative Randomization." NeurIPS, 2021.



**Summary Of The Paper:**

This paper proposes Signed Supermask,  an extension of the original Supermask work (Zhou2019) for finding more efficient untrained subnetworks. Instead of learning a binary mask, Signed Supermask claims and shows that adding another dimension -1 to the masks leads to higher sparsity with higher accuracy. The main contribution of this paper is the introduction of weight flipping, as an improvement over the existing approaches. The method is simple and effective. The empirical experiments validate the effectiveness of the proposal.

**Summary Of The Review:**

While the performance achieved by untrained NN is interesting, the motivation and experiments need more work.

---

> ### Author Response · Authors · 2021-11-16
> **Author Response**
>
> The authors thank the reviewer for the constructive criticism.
>
> **Argument: Training Time / Efficiency** \
> The authors agree that signed Supermasks do not speed up training. However, this is not the point of our work. The focus of this work is to find the smallest possible subnetwork that still performs well while not changing the weight values. We argue that signed Supermasks uncover a new generation of extremely small subnetworks (with weights frozen at initialization) that are much easier to interpret in future work. \
> The reviewer is correct that we do train the neural network, however, the weight values are not adapted, hence *untrained*. The reviewer is further correct, that the gradient is dense. In our case, this is necessary to "switch on" weights, that were previously masked. It is an interesting idea to apply sparse gradient methods to a signed supermask-like approach. \
> The benefits do not lay in the practicality of our approach, just like in the original work of Supermasks [1], later [2] and the Lottery Ticket Hypothesis [3]. Note that none of those works reported on training time, which can lead to the conclusion that their methods do not train a neural network more efficiently. Rather, the benefits are in a greater understanding of what is actually important in a neural network. Our work has uncovered far smaller subnetworks than previous work, which simplifies interpretation in future work. We have shown that our ELU(S) initialization with a signed constant approach is sufficient to initialize signed Supermask networks and even their dense counterparts. Our work supports previous literature in the notion that the connection itself might be more important than its value and found that the uncovered subnetworks are similar throughout different runs. The benefit of this work is that it kicks-off and enables many new approaches to understand and interpret neural networks in general by providing the smallest possible subnetworks with very good performance.
>
> **Argument: Literature** \
> We thank the reviewer for the pointers to this literature. We have added Table 14 in Appendix C that compares 16 related works with our method. While dynamic pruning is an interesting concept, it differs from a supermask-like approach in the sense that weight values stay *constant*. We still realize very sparse networks. The focus is also not to train the networks more efficiently, yet, but to understand what is needed in a good network to train a smaller version in the future. Furthermore, efficiency is gained after training. Mocanu et al. [4] start with a quite large initial neural network, which results in a much higher probability to find a good subnetwork at initialization. The final total number of remaining weights is still much higher compared to our signed Supermasks. For their CNN experiments, it seems that they only prune the fully-connected layers. They do not report on training time, which leads to the suspicion that their method is not faster (than normal training) as well. Evci et al. [5] apply the Lottery Ticker Hypothesis "online" during training, which is an intriguing idea. They also start with a very large architecture (>26M parameters). The same is true for Liu et al. [6].
>
> **Argument: Related work** \
> We thank the reviewer for pointing us towards those works. We completely agree that those works should be included and we updated the respective sections in our work. We would like to point out, that neither Diffenderfer \& Kailkhura [7] nor Chijiwa [8] report on their method's efficiency during training for their experiments.
>
> **Arguments: Experiments on initialization and threshold**
> - Table 12 in Appendix C shows results of ELUS, He [9] and Xavier [10] for all investigated CNN architectures while Appendices D and E investigates the influence of weight and mask initialization. We added Appendix F, in which we present results on the control of the pruning rate and extreme pruning. We have made the references to those sections more clear in the main text and thank the reviewer for pointing out this weak point.
> - The authors agree that the threshold parameter is a hyperparameter that needs to be tuned. Preliminary experiments have shown that there is an influence on performance but on the other hand is not that interesting from an interpretation perspective: it needs to be chosen carefully but there's not much more to it. We have focused our experiments on different initialization methods and distributions for weights and mask and the limits of extreme pruning
>
> [1]  https://arxiv.org/abs/1905.01067 \
> [2]  https://arxiv.org/abs/1911.13299 \
> [3]  https://arxiv.org/abs/1803.03635 \
> [4]  https://www.nature.com/articles/s41467-018-04316-3.pdf \
> [5]  https://arxiv.org/abs/1911.11134 \
> [6]  https://arxiv.org/abs/2106.10404 \
> [7]  https://arxiv.org/abs/2103.09377 \
> [8]  https://arxiv.org/abs/2106.09269 \
> [9]  https://arxiv.org/abs/1502.01852 \
> [10]  https://proceedings.mlr.press/v9/glorot10a/glorot10a.pdf

---

### Official Review · Reviewer_UFKs · 2021-11-01

**Correctness:** 2
**Technical Novelty And Significance:** 2
**Empirical Novelty And Significance:** 2
**Recommendation:** 5
**Confidence:** 4

**Main Review:**

Previous studies on supermask are interesting because they provide insights that even untrained models have chances to achieve reasonable model accuracy only if we can find such supermask in advance. This reviewer is not sure whether minor improvements over the original supermask idea are critical if a large amount of training is still required.

If the authors can reveal that the model accuracy can be significantly improved even for untrained models compared to previous supermask, it would be helpful to understand the inherent characteristics of the neural networks initialization. But as shown in Table 1, compared to the work by Zhou et al, even test accuracy is slightly degraded even with additional "-1" mask while 'normal training' is still required. The original idea of Supermask is meaningful because it shows a new insight that untrained model has already a sub-network of reasonable accuracy. If the authors want to claim that we can generate a new Supermask generation method, it would not produce additionally new insight. Such concern is obvious for residual networks. Without training batch norm parameters, it is unavoidable to see noticeable accuracy degradation. Then, batch-norm hyperparameters are known to be affine hyperparameters that are superior to normal weights in terms of expressive power. Since the existence of Supermask is already known, the focus of new research in Supermask would need to be finding Supermask efficiently with minimal efforts. Otherwise, Supermask would not be practical in the field. Unfortunately, as indicated in Table 1, training time to find Supermask is quite slow. Then, why we don't just perform usual pruning method to achieve better model accuracy?

Comparison on the previous works is missing. At least, this paper needs to include "What's Hidden in a Randomly Weighted Neural Network?" in CVPR 2020.

**Summary Of The Paper:**

This paper discusses the signed supermask that can improve the model accuracy of untrained neural networks significantly while enhancing sparsity compared to the original supermask idea. The authors introduce "-1" as an additional mask value to enable flipping a sign of an initialized weight and suggest related activation functions and fixed threshold hyperparameters to achieve sparse networks. Analysis on sparsity and corresponding accuracy is given for various CNN models including ResNet models. The role of batch normalization for large models is also studied.

**Summary Of The Review:**

Since this paper introduces only marginal improvements over original Supermask work, this reviewer cannot find significantly new insights from this paper. Supermask needs to involve much higher model accuracy (than previous works) or minimal efforts to be computed.

---

> ### Author Response · Authors · 2021-11-16
> **Author Response**
>
> The authors thank the reviewer for the constructive criticism.
>
> **Argument: Only minor improvements**
> - While the authors fully agree with the reviewer, that signed Supermasks are an evolution of the groundbreaking idea of Supermasks by Zhou et al. [1], we do see significant contributions: we are able to uncover subnetworks that are consistently much smaller and perform better compared to Zhou et al. [1] and Ramanujan et al. [2]. When analyzing the internal structure of a pruned neural network, smaller subnetworks will always be easier to work with and interpret, which underlines the importance of signed Supermasks. \
> The work furthermore includes a novel initialization scheme with theoretical justification for the ELU activation function, which can be used for any neural network (not only signed Supermasks). Our experiments show that this scheme is beneficial for signed Supermasks as well.
> - We further want to highlight, that the focus of our work is not on providing a faster training method. However, we presume that signed Supermasks are considerably faster to train than the Supermasks of Zhou et al. [1] (since we have to draw random variables in each step) and Ramanujan et al. [2] (because no sorting of values is necessary). Initial experiments on our side confirm that the approach of Ramanujan et al. [2] takes much longer compared to (signed) Supermasks. Since the experimental setup was prototypical, we did not show this in our work. Note that even the original work on the Lottery Ticket Hypothesis [3] states that iterative magnitude pruning takes much longer to train. We reported the training time statistics to be as transparent with the proposed idea as possible.
> - Our main focus is to drive a general understanding of neural networks forward to enable future work to build smaller networks (with lower training times) from the start. Signed Supermasks lay the foundation of this by maximizing pruning rate without adapting weight parameters at all. To us, this finding is fascinating in itself. To show, that they can perform even better than the baselines is even more intriguing. Simply pruning a trained network during or after training does not offer the same insights, otherwise we can question all the literature on the Lottery Ticket Hypothesis, in the sense that this literature is contributing to find and understand subnetworks from initialization. The big goal is to be able to train uncovered subnetworks from the start and understand why those perform better than others.
>
> **Comment on missing related Work**
> - We quote the paper "Deconstructing lottery tickets: Zeros, signs, and the supermask" by Zhou et al. (published NeurIPS 2019) [1] many times and see our work as an evolutionary development of their work. The results of FCN, and Conv 2,4,6,8 are directly compared to their results. The authors are not sure if the reviewer refers to the same work, as we were unable to find a paper with the same title published in CVPR 2020.
>
> [1]  https://arxiv.org/abs/1905.01067 \
> [2]  https://arxiv.org/abs/1911.13299 \
> [3]  https://arxiv.org/abs/1803.03635

---

> > ### Author Response · Authors · 2021-12-09
> > **Additional Author Response**
> >
> > The authors noted that the reviewer changed the comment about the absence of the paper "Deconstructing Lottery Tickets" to the absence of the paper "What's Hidden in a Randomly Weighted Neural Network?" in his comment about missing work.
> >
> > Therefore, we would like to add to our original answer, that the paper mentioned by the reviewer "What's Hidden in a Randomly Weighted Neural Network?" by Ramanujan et al. is cited many times in our work as well. We compare our experimental results with their edge-popup algorithm in the main part of the paper. However, we (so far) cite the arxiv-version of the article and thank the reviewer for implicitly pointing us onto that. In the final version of our work, we will change the citation to the CVPR-version accordingly.

---

### Official Review · Reviewer_sRRw · 2021-11-02

**Correctness:** 3
**Technical Novelty And Significance:** 3
**Empirical Novelty And Significance:** 3
**Recommendation:** 8
**Confidence:** 5

**Main Review:**

**Strength:**

- This paper is well written and the proposed method is clearly described.
- The proposed method is fairly novel, and the resulting high performance and sparsity level of the Conv 2,4,6,8 models are inspiring.

**Weakness:**

- The paper motivates the signed supermask as a way to improve the interpretability of the trained model. While this would be very interesting, I find that the supporting interpretability analyses to be lacking in substance. The mask visualizations are limited to the first layer of a FC network on MNIST, and cannot be directly extended to more layers or other architectures. The observations of layer-wise pruning ratios are interesting, but are not new and have been well studied in the pruning literature.

- Another motivation is in terms of efficiency due to sparsity and compression of the final trained model. To this point, I think it is more appropriate to compare performance to other tertiary network methods, as the supermask baselines have much less capacity due to maintaining the sign at initialization. Is signed supermask a competitive method for training tertiary networks? How does it compare to other tertiary methods in terms of the performance and efficiency tradeoff? Testing the limits of how barebones can a network be is an interesting question, and it may very well be the case that the signed supermask pushes this limit, but it needs to be made clear through a quantitative comparison to similar methods rather than just a discussion.

- Comparisons to the other supermask baselines are missing in the ResNet models for some reason.


**Summary Of The Paper:**

This paper introduces "signed supermasks", which builds on top of the supermasks line of work of finding binary masks on untrained networks that result in good performance. This work extends the original supermask by adding the ability to flip the sign of the weights. The proposed method learns parameters of a mask, and converts the mask parameters into a tertiary mask through the use of two thresholds. The paper further proposes to use ELU activation function and a ELUS initialization scheme that is more tailored to supermask training. Performance is compared against fully-trained models and prior work on supermasks on MNIST, CIFAR10, and CIFAR100, and the proposed method shows competitive results.

**Summary Of The Review:**

This paper presents an interesting new approach of training tertiary neural networks with high sparsity levels. The paper is well written and several of the results look promising and would be of interest to the community. However, I take some issue with the motivations of this paper. As a method that is proposed to produce insights on neural network training and enable interpretablility of networks, I find a lack of new insights provided in the paper. As a method that improves efficiency and compressibility, I find it to be very intriguing and promising, but the paper currently lacks the appropriate performance comparisons with other tertiary neural networks to fully convince me of its advantages.

### Update after rebuttal

Thank you to the author for the update. I agree that there is value in training the most bare-boned network that we can. Even though not all the value has been demonstrated in the present work, I think it will be useful in many, perhaps unexpected ways in the future. Although this is not really discussed, I'm particularly interested in how this form of training can act as a particular form of regularization that is orthogonal to most other forms. I'll raise my score from 5 to 8 and encourage the authors to continue in this direction.

---

> ### Author Response · Authors · 2021-11-16
> **Author Response**
>
> The authors thank the reviewer for the constructive criticism.
>
> **Argument: Missing novel interpretability measures**
> - The authors fully agree with the reviewer in the sense that our work does not propose novel interpretability analyses/measures. That being said, we believe that signed Supermasks enable future work to interpret neural networks much easier. Compared to pruning, (signed) Supermasks do not adapt the weight values, which delivers a more fundamental understanding to neural network functionality. This is discussed at length in the original Lottery Ticket Hypothesis paper by Frankle \& Carbin [1]. Supermasks take this approach merely a step further, while signed Supermasks reach even smaller size while performing consistently better. \
> By using signed constants, we take our eyes off the weight values and rather focus our attention on the structure of the neural network, which is even similar throughout different runs according to the measures applied in this work. For example, a straight-forward starting point would be to analyze the first layer mask and combine these analyses with the used features to use signed Supermasks as feature selectors or feature importance measures. It is crucial for this to get the smallest possible subnetwork while at least maintaining the performance of the original network, which gives signed Supermask an advantage over the original Supermasks. This field, however, far exceeds the scope of this work.
>
>
> **Argument: Comparison to TNNs missing**
> - The authors support the statement of the reviewer, that signed Supermasks inhibit similarities with TNNs, which is discussed in the paper as well. We have added a table (Table 14) in Appendix C, that lists 16 broadly related works, including binarized and ternarized NNs. Please note, that a direct performance comparison is still not possible as the starting point for all those works varies immensely in terms of their method and its complexity, the initial network architectures and usage of additional layers such as batch normalization or dropout. \
> That being said, signed Supermasks are as similar to TNNs as Supermasks [2,3] and even the Lottery Ticket Hypothesis [1] to binarized neural networks. We decided to stick to the LTH/Supermask-related literature in the main part of paper. Since the FCN and Conv 2,4,6,8 are not investigated in the TNN literature at all many more experiments with different architectures would be required to make an adequate comparison. We call future research to exactly identify similarities and differences in both strands of literature, such that both can learn from each others thinking.
>
> [1]  https://arxiv.org/abs/1803.03635 \
> [2]  https://arxiv.org/abs/1905.01067 \
> [3]  https://arxiv.org/abs/1911.13299

---

> > ### Comment · Reviewer_sRRw · 2021-11-29
> > **Thank you for your response**
> >
> > I appreciate the author's response. I think there is value to this work, though I still feel that the present paper has a lot of room for improvement in providing concrete demonstrations of this value. Original supermasks are certainly not appropriate as the only performance baselines, but I suspect there may be other benefits of this type of approaches beyond what is measured in the paper. I would like to give this work the benefit of the doubt, and have raised my score accordingly. Nonetheless, I would encourage authors to show concrete new insights derived from the signed supermask model as that would further strengthen this work.

---

### Official Review · Reviewer_AEMP · 2021-11-10

**Correctness:** 3
**Technical Novelty And Significance:** 2
**Empirical Novelty And Significance:** 2
**Recommendation:** 6
**Confidence:** 4

**Main Review:**

Firstly, this paper is written clearly and easy to follow. The main technical contributions are clear. Performance improvements over (Zhou et al., 2019) and (Ramanujan et al., 2019) are clear.

However,  there are several concerns I have for this work.

1. I think more detailed ablation study is needed for the experiments. For example, it is imaginable that the threshold tau will have an influence on the number of remaining parameters but the corresponding experiments are lacking. Having some will be nice. Also, the new training technique is always coupled with the new initialization method. Ablation study to decouple the effect of these two contributions will be good for the readers to understand the which one is more important. This is crucial because in (Ramanujan et al., 2019) the authors also investigated a certain form of scaling of the parameters which is found to improve the performance significantly.

2. In the abstract, the authors posed this work as tackling the issue of difficult interpretation and training/usage in real-world applications of highly overparameterized networks due to their size. However, I don't think the proposed method save anything during training, e.g., memory, computations (in FLOPs) or time because it uses latent weights.

Minor point:
 - In table 2, should the baseline accuracy for conv 8 be 82+% instead of 72+%?

**Summary Of The Paper:**

This work extends to (Zhou et al., 2019) and (Ramanujan et al., 2019) to allow sign flipping in the supermasks, without updating the magnitudes of weights. The main technical contributions are the new thresholding-based training method and the new weight initialization scheme that considers the supermasks. Empirical results show better performance on small conv nets and also residual networks on CIFAR-10/100 datasets.

**Summary Of The Review:**

Overall I think this work proposes an interesting extension to previous supermask works but lack necessary experiments, making the results less convincing.

---

> ### Author Response · Authors · 2021-11-16
> **Author Response**
>
> The authors thank the reviewer for the constructive criticism.
>
> **Argument: More experiments needed**
> - We thank the reviewer for pointing out that the experimental results shown in the Appendix are not mentioned clear enough in the main paper and changed it accordingly. We show experimental results for He [2] and Xavier [3] on signed Supermasks and the baselines, as well as a uniform and normal weight distribution compared to the signed constant approach. Further, we investigated the influence of mask initialization in the Appendices C, D and E. Moreover, we added another section in the Appendix (Appendix F), investigating the control of the pruning rate and extreme pruning. Preliminary experiments on the choice of the threshold parameters have shown that there is an influence on performance but on the other hand is not that interesting from an interpretation perspective: it needs to be chosen carefully but there's not much more to it. We focused, as the reviewer rightly points out, on the more important topic of initialization method and scheme.
>
> **Argument: Save during training**
>
> - We fully agree with the reviewer in the sense that signed Supermasks require longer training time, as reported in Tables 1 and 3 for FCN and Conv architectures. However, we want to highlight that none of the previous literature on Supermasks even report on training time, which can lead to the speculation that those require more training time as well. In fact, our preliminary experiments have shown that an implementation of Ramanujan et al [1] requires a multiple of training time, as hinted in the first paragraph of Section 3.2. Further note, that the authors of the Lottery Ticket Hypothesis state a similar train of thought in their work. Our focus, however, is not a practical out-of-the-box tool as of yet. The goal was much more to investigate the fundamentals of neural network functionality. For the authors, it is still fascinating to see how small, yet performant the found subnetworks can become.
> - The authors further agree, that this work is not providing novel techniques to interpret neural networks. However, this is not the focus of this work. We argue that signed Supermasks (compared to Supermasks) uncover a new generation of extremely small subnetworks (with weights frozen at initialization) that are much easier to interpret in future work. This is a big task on its own and is therefore not feasible to do in this work as well. The found subnetworks are even similar throughout different runs according to the measures used in our work. We hinted on the difficulty of comparing the trained masks in our work but this is another very interesting field of research that is enabled by signed Supermasks.
>
> *Minor point*: The reported results in Table 2 are correct.
>
> [1]  https://arxiv.org/abs/1911.13299 \
> [2]  https://arxiv.org/abs/1502.01852 \
> [3]  https://proceedings.mlr.press/v9/glorot10a/glorot10a.pdf

---

### Decision · Program_Chairs · 2022-01-20

**Decision:**

Accept (Poster)

**Comment:**

This paper builds on previous work on supermasks. It  proposes to replace binary masks by a signed supermask, i.e. a trainable, trashold-based mask that can take values from {-1,0,1}. This change (in combination with the use of ELUS activation functions and an ELUS specific initialization strategy) leads to a significantly higher pruning rate while keeping competitive performance  in comparison to baseline models.

Most reviewers agreed that the paper is well written and that the proposed approach and the experimental findings are interesting. The motivation to improve interpretability was commonly perceived as misleading. Another  downside that was mentioned is the training time/efficiency. This however, should not be taken too much into account since the work focusses on finding the smallest possible subnetwork that still performs well (without changing the weight values) and- in line with work on the lottery hypothesis-  aims at understanding more about the structure of the „winning tickets“ which is interesting for itself. The paper therefore should be accepted.